# An essential role of acetylcholine-glutamate synergy at habenular synapses in nicotine dependence

Silke Frahm[1†‡], Beatriz Antolin-Fontes[1,2†], Andreas Görlich[2†], Johannes-Friedrich Zander[3], Gudrun Ahnert-Hilger[3], Ines Ibañez-Tallon[1,2*]

[1]Molecular Neurobiology Group, Max Delbrück Center for Molecular Medicine, Berlin, Germany; [2]Laboratory of Molecular Biology, The Rockefeller University, New York, United States; [3]Institute for Integrative Neuroanatomy, Charité - Universitätsmedizin Berlin, Berlin, Germany

**Abstract** A great deal of interest has been focused recently on the habenula and its critical role in aversion, negative-reward and drug dependence. Using a conditional mouse model of the ACh-synthesizing enzyme choline acetyltransferase (*Chat*), we report that local elimination of acetylcholine (ACh) in medial habenula (MHb) neurons alters glutamate corelease and presynaptic facilitation. Electron microscopy and immuno-isolation analyses revealed colocalization of ACh and glutamate vesicular transporters in synaptic vesicles (SVs) in the central IPN. Glutamate reuptake in SVs prepared from the IPN was increased by ACh, indicating vesicular synergy. Mice lacking CHAT in habenular neurons were insensitive to nicotine-conditioned reward and withdrawal. These data demonstrate that ACh controls the quantal size and release frequency of glutamate at habenular synapses, and suggest that the synergistic functions of ACh and glutamate may be generally important for modulation of cholinergic circuit function and behavior.

*For correspondence: iibanez@rockefeller.edu

†These authors contributed equally to this work

Present address: ‡Institute of Pharmacology/CCR, Charité - Universitätsmedizin Berlin, Berlin, Germany

Competing interests: The authors declare that no competing interests exist.

## Introduction

Acetylcholine (ACh) was first described in the heart muscle (*Loewi, 1921*), and later in the peripheral nervous system, as a fast acting neurotransmitter at the neuromuscular junction (*Bennett, 2000*). In the central nervous system (CNS), however, evidence supports the hypothesis that ACh acts by volume transmission and that slowly changing levels of extracellular ACh mediate arousal states contributing to attention, sleep, learning and memory (*Dani and Bertrand, 2007*; *Everitt and Robbins, 1997*; *Mesulam et al., 1983*; *Picciotto et al., 2012*; *Ren et al., 2011*; *Sarter et al., 2009*). Extracellular ACh levels are limited by acetylcholinesterase (AChE), which cleaves ACh into choline and acetyl-coA (*Rosenberry, 1975*). Cholinergic projection neurons in the medial habenula (MHb) synapse in the interpeduncular nucleus (IPN), which contains extremely high levels of AChE (*Flumerfelt and Contestabile, 1982*). At synapses with a high concentration of AChE, ACh is so quickly degraded that a single molecule cannot activate a second receptor (*Kuffler and Yoshikami, 1975*). Since both nicotinic (nAChRs) and muscarinic (mAChRs) acetylcholine receptors are often localized extrasynaptically on dendrites and somata, and presynaptically at axonal terminals (*De-Miguel and Fuxe, 2012*; *Descarries et al., 1997*; *Role and Berg, 1996*), it is thought that ACh volume transmission in the IPN requires high frequency stimulation of habenular neurons (*Ren et al., 2011*). The release of ACh from habenular terminals is consistent with genetic studies demonstrating altered responses to nicotine addiction as a consequence of mutations in nicotinic receptors that are enriched in the MHb-IPN (*Antolin-Fontes et al., 2015*; *Fowler et al., 2011*; *Jackson et al., 2010*; *Salas et al., 2009*).

**eLife digest** Neuroscientists are making progress in understanding the brain regions and neural circuits that are involved in reward and addiction. One such region, called the habenula, is found near the center of the brain and sends nerves to another brain region called the interpeduncular nucleus (or IPN for short); this creates a neural circuit that is important for the brain's responses to nicotine.

The habenular-IPN circuit is rich in receptors for a neurotransmitter called acetylcholine. These receptors are also activated by nicotine, the addictive component of tobacco. Neurotransmitters are chemicals that transmit a signal from one nerve to another. These chemicals are packaged into small structures called vesicles, which are found at nerve endings. When a nerve impulse reaches the end of a nerve, it triggers the release of a vesicle's contents into the gap (or 'synapse') between nerve cells. The released neurotransmitters can then bind to receptors on the neighboring nerve cells before being cleared away.

The nicotinic acetylcholine receptors in the habenula-IPN circuit are associated with nicotine dependence in mice and humans. Frahm, Antolin-Fontes, Görlich et al. have now investigated what happens when the gene encoding the enzyme that makes acetylcholine is removed from habenular nerves in mice. The experiments revealed that these mutant mice become insensitive to the rewarding properties of nicotine and are protected from the effects of its withdrawal, following long-term exposure. The loss of acetylcholine from habenular nerves was also found to reduce the generation of small electrical currents in the nerves of the IPN. These currents are generated by another neurotransmitter called glutamate. If brain slices from normal mice are infused with acetylcholine or nicotine, these currents become more frequent. This response is not seen in brains of the mutant mice suggesting that acetylcholine helps the release of glutamate from habenular nerve endings.

Frahm, Antolin-Fontes, Görlich et al. then found that the proteins that transport glutamate and acetylcholine into synaptic vesicles are found at the same sites in nerve endings in the IPN. Further experiments showed that acetylcholine also increases the reuptake of glutamate into synaptic vesicles and controls the amount and frequency of glutamate released at habenular synapses.

These results thus reveal how acetylcholine mediates the effects of nicotine on the brain, in part by regulating the uptake and release of glutamate by habenular nerve endings, identifying a new mechanism important for nicotine dependence. Since many acetylcholine-releasing nerve cells also release glutamate, a future challenge will be to investigate whether the interaction between these two neurotransmitters is important for other processes that rely on acetylcholine, such as memory and thought.

Although a clear role for ACh in volume transmission and for nAChRs in nicotine dependence has been established, several observations suggest that ACh may have additional functions in cholinergic neurons. For example, it has been demonstrated in several systems that neurotransmitters can be coreleased (*El Mestikawy et al., 2011*; *Gras et al., 2008*; *Guzman et al., 2011*; *Hnasko et al., 2010*; *Hnasko and Edwards, 2012*; *Ren et al., 2011*; *Shabel et al., 2014*), and that cooperation between vesicular neurotransmitter transporters located in the same synaptic vesicle (SV) can reciprocally increase the packaging of their respective neurotransmitters into SVs, a process termed vesicular synergy (*El Mestikawy et al., 2011*; *Gras et al., 2008*; *Hnasko et al., 2010*). Furthermore, infusion of glutamate receptor antagonists into the IPN results in decreased nicotine intake (*Fowler et al., 2011*) and withdrawal (*Zhao-Shea et al., 2013*) suggesting that glutamate-mediated fast synaptic transmission is also important in nicotine addiction. Given evidence that both glutamate and ACh play important roles at habenular-IPN synapses, and that vesicular synergy contributes to the physiology and function of other critical CNS circuits, we were interested in examining the interactions between these transmitter systems in habenular neurons and in determining their contributions to nicotine dependence.

To investigate the interactions between glutamate and ACh in the MHb-IPN circuit, we locally eliminated ACh at this synapse by genetic deletion of the ACh-synthesizing enzyme choline acetyltransferase (*Chat*) in MHb neurons in conditional knockout (cKO) mice. Immunogold electron

microscopy revealed that ACh and glutamate vesicular transporters colocalize in a significant fraction of SVs in the central IPN (IPC). Patch clamp recordings showed that glutamatergic miniature excitatory postsynaptic currents (mEPSCs) were smaller in IPC neurons of ChAT-cKO, but unchanged in neurons of the lateral IPN (IPL) which receives non-cholinergic input, indicating that glutamate release is reduced in the absence of ACh. Upon ACh or nicotine superfusion, wild-type (wt) IPC neurons displayed an increased frequency of mEPSC. In ChAT-cKO slices this response was not observed, indicating that presynaptic facilitation of glutamate release is impaired. Direct measurements of glutamate reuptake into IPN SVs in the presence and absence of ACh demonstrated vesicular synergy. Behaviorally, ChAT-cKO mice were insensitive to the rewarding properties of nicotine and displayed no withdrawal signs after cessation of chronic nicotine treatment. Our results, therefore, establish an essential role for ACh corelease with glutamate in habenular neuron function and reveal an additional mechanism that may play an important role in nicotine dependence.

## Results

### Conditional gene deletion of *Chat* in cholinergic habenular neurons

CHAT is the only enzyme that synthesizes ACh, and it is expressed at the neuromuscular junction and in several brain areas including the MHb, basal forebrain (BF), laterodorsal tegmental nucleus (LDTg), third cranial nerve (3N) and nucleus of the solitary tract (NTS) (*Figure 1A*). Null mutant mice for *Chat* obtained by crossing a floxed allele of the *Chat* gene (*Chat$^{flox/flox}$*) to β-*actin*-Cre transgenic mice die at birth (*Misgeld et al., 2002*). Therefore to elucidate the contribution of cholinergic transmission to the function of the MHb-IPN pathway, we sought to conditionally delete *Chat* in habenular neurons. To drive Cre-recombinase activity specifically to MHb cholinergic neurons we analyzed their translational profile (*Gorlich et al., 2013*) and selected the *Kiaa1107-Cre* mouse BAC transgenic line for its specific pattern of expression in the MHb (GENSAT, www.gensat.org; *Figure 1A*). *Kiaa1107-Cre* mice were crossed to reporter *Gt(ROSA)26Sor$^{tm1(EYFP)Cos}$* mice (*Figure 1B*) to verify that EYFP expression resulting from Cre-recombinase activity was achieved in MHb neuron somata and habenular axonal projections in the IPN (*Figure 1C*). Double immunostaining with CHAT and EYFP antibodies in *Kiaa1107-Cre* mice crossed to reporter *Gt(ROSA)26Sor$^{tm1(EYFP)Cos}$* mice (*Figure 1D*) demonstrated that 99% (1912 of 1933) of CHAT positive neurons in the MHb are positive for the EYFP reporter. In contrast, CHAT populations in striatum, PPTg and LDTg show extremely low expression of the EYFP reporter (0.5 to 1.3% of CHAT cells) (*Figure 1E*). These results establish that the *Kiaa1107-Cre* line specifically targets the cholinergic population of habenular neurons without affecting other cholinergic neurons.

Previous studies have demonstrated that a conditional allele of the *Chat* gene (*Chat$^{flox/flox}$*), where exons 3 and 4 are flanked by loxP sites, generates a *Chat$^{-/-}$* allele when these exons are excised by Cre-recombinase (*Misgeld et al., 2002*) (*Figure 2A*). Thus, to remove the enzyme from habenular neurons, we employed *Kiaa1107-Cre* mice to drive conditional deletion of the CHAT enzyme in habenular neurons (*Figure 2A*). Western blot analyses of habenular and IPN brain extracts revealed absence of CHAT in double positive mice for *Kiaa1107-Cre* and *Chat$^{flox/flox}$* (*Figure 2B*), hereafter referred to as ChAT-cKO mice. Immunohistochemical analyses of brain sections clearly showed that CHAT immunoreactivity was absent in the MHb, fasciculus retroflexus (fr) and IPN in ChAT-cKO mice (*Figure 2C–D*). To assess the penetrance of the driver Cre-line we quantified the number of neurons that remained positive for CHAT in ChAT-cKO mice across different cholinergic areas (*Figure 2E, F*). This analysis showed that only 0.3% habenular neurons in ChAT-cKO mice retained their immunoreactivity to CHAT, while the number of CHAT positive neurons in striatum, PPTg and LDTg were comparable in wt and ChAT-cKO mice (*Figure 2F*). ChAT-cKO and wt mice also displayed comparable immunoreactivity for CHAT in other cholinergic brain areas including BF and third cranial nerve (3N) (*Figure 2G*). These data show that the *Kiaa1107-Cre* strain drives Cre-recombination of the *Chat* conditional allele in 99.7% of habenular cholinergic neurons, and that it can be used to specifically delete *Chat* only from habenular neurons without perturbing other cholinergic sources in the brain. To determine whether *Chat* excision occurred during the early stages of habenular development (*Quina et al., 2009*), we analyzed the expression of CHAT in wt and ChAT-cKO at early postnatal ages and detected the onset of *Chat* Cre-mediated excision by *Kiaa1107-Cre* between postnatal days P6 and P7 (*Figure 3*). Taken together, these data show that this genetic manipulation efficiently

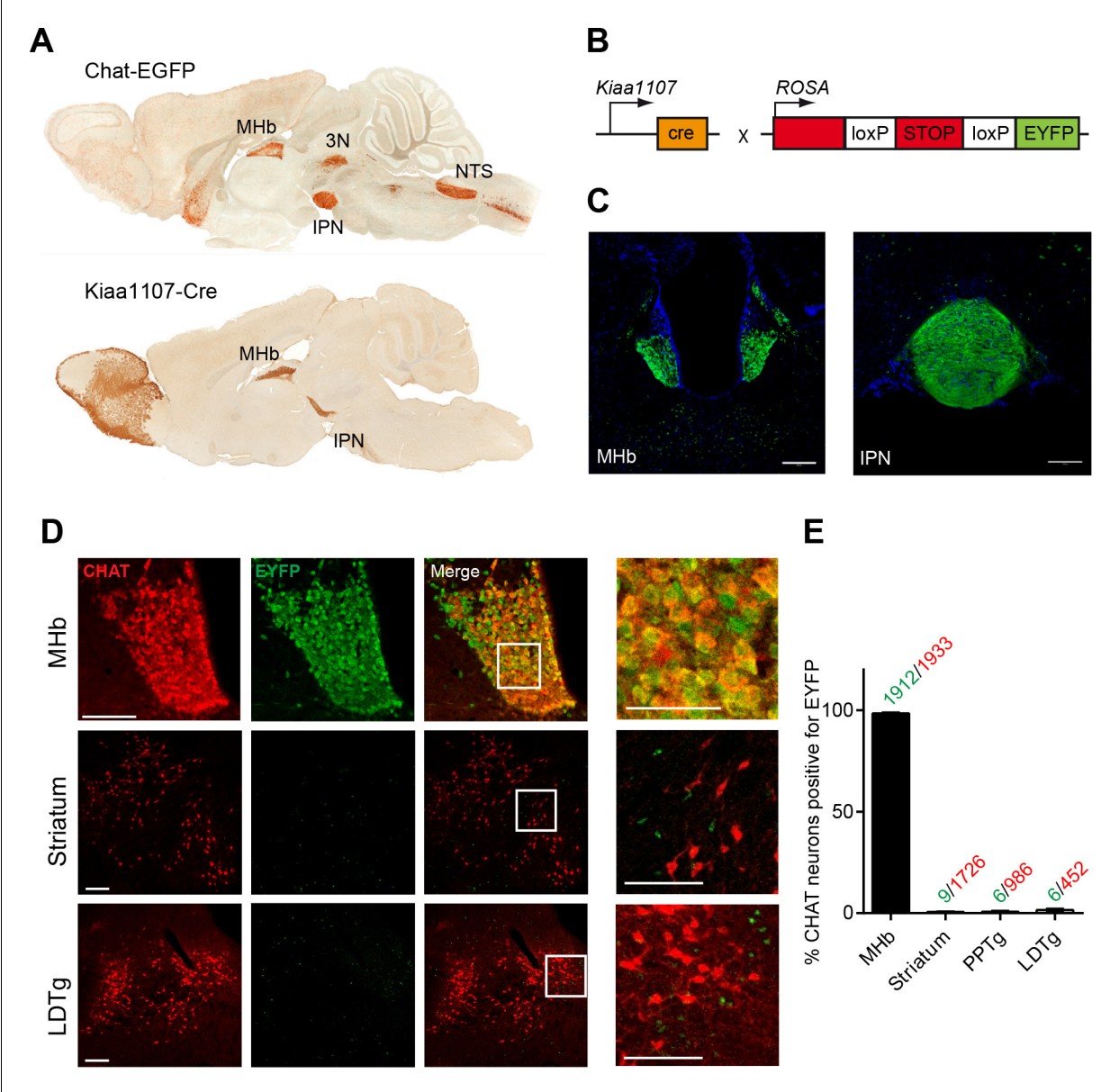

**Figure 1.** Analysis of the Cre driver line *Kiaa1107-Cre* in cholinergic neurons. (**A**) Sagittal images from GENSAT corresponding to mouse BAC transgenic lines: *Chat-EGFP* founder GH293 and *Kiaa1107-Cre* founder KJ227. *Chat-EGFP* mice show EGFP expression in cholinergic areas including MHb, habenular projections to the interpeduncular nucleus (IPN), the laterodorsal tegmentum (LDTg), third cranial nerve (3N), basal forebrain (BF), and nucleus of the solitary tract (NTS). *Kiaa1107-Cre* mice show Cre-recombinase expression in the MHb and axonal projections in the IPN. (**B**) Mouse breeding scheme of the Cre-recombinase *Kiaa1107-Cre* transgenic line crossed with the Cre-dependent reporter line *Gt(ROSA)26Sor^{tm1(EYFP)Cos}* to visualize Cre-recombinase activity. (**C**) Cre-dependent EYFP-expression driven by *Kiaa1107-Cre* was observed in the ventral two-thirds of the MHb and in the axonal habenular projections to the central IPN. Scale bars: 200 μm. (**D**) Double immunostaining analyses with CHAT (red) and EYFP (green) antibodies in cholinergic brain areas of *Kiaa1107-Cre* crossed to *Gt(ROSA)26Sor^{tm1(EYFP)Cos}* mice. High magnifications of the indicated square areas are shown on the right column. Neurons in the ventral part of the MHb are double positive, while CHAT and EYFP label different cells in striatum and LDTg. Scale bars: MHb 100 μm, inset 50 μm and striatum, LDTg 200 μm, insets 100 μm. (**E**) Quantification of CHAT positive neurons expressing EYFP in MHb, striatum and tegmental cholinergic areas (LDTg, laterodorsal tegmentum; PPTg, pedunculopontine nucleus). The number of double positive cells per brain area is indicated above each bar (n=8 to 16 sections from three different mice). MHb: Medial habenula.

and selectively eliminates *Chat* in cholinergic habenular neurons, and that it does so after formation of the MHb/IPN circuitry. The ChAT-cKO mouse, therefore, is a useful model in which to test the consequences of selectively removing one neurotransmitter in a specific axonal tract.

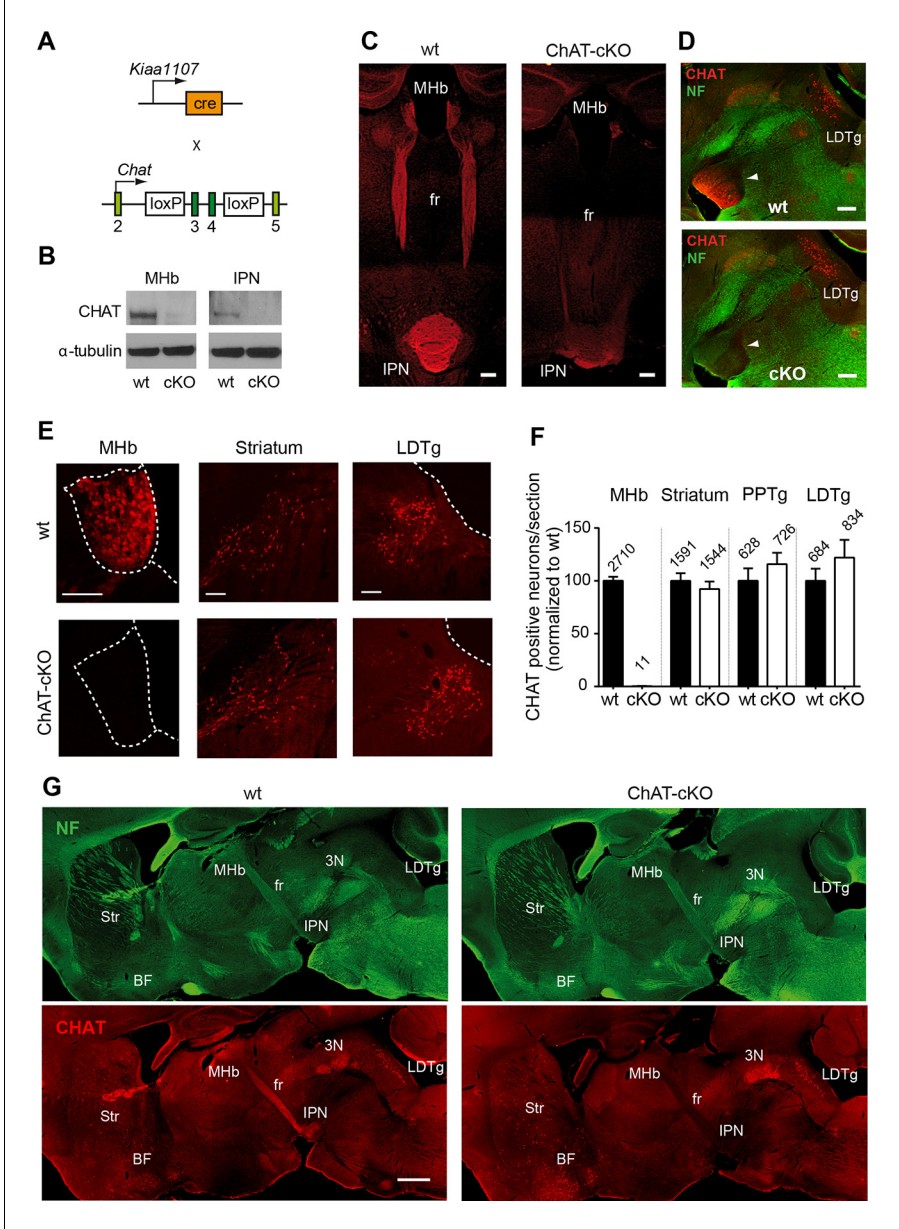

**Figure 2.** Conditional gene deletion of *Chat* in cholinergic neurons of the MHb. (**A**) Mouse breeding scheme of the Cre-recombinase *Kiaa1107* line crossed to *Chat*flox/flox mice to generate ChAT-cKO mice with conditional gene deletion of *Chat* in habenular neurons. (**B**) Western blot analysis with CHAT and α-tubulin antibodies in MHb and IPN extracts from wt and ChAT-cKO mouse brains. (**C**) Angled sections of the midbrain immunostained for CHAT (red). In wt mice (left panel), CHAT is highly expressed in MHb neurons, along their axons in the fasciculus retroflexus (fr) and in their axonal terminals in the IPN. In ChAT-cKO mice (right panel), CHAT immunoreactivity is no longer detected in the MHb-fr-IPN tract. Scale bars: 300 μm. (**D**) Sagittal sections of the midbrain immunostained for CHAT (red) and neurofilament (green). CHAT immunoreactivity is strong in the IPN (arrowhead) of wt mice (upper panel), and absent in ChAT-cKO brains (lower panel) while the laterodorsal tegmental nucleus (LDTg) - an efferent target of the IPN - and adjacent third cranial nerves (3N) show similar CHAT expression in wt and ChAT-cKO mice. Scale bars: 300 μm. (**E**) Immunostaining analyses in wt and ChAT-cKO show that CHAT immunoreactivity (red) is no longer detected in the MHb of ChAT-cKO mice. Wt and ChAT-cKO mice show similar CHAT immunoreactivity in striatum and LDTg. Scale bars: MHb and striatum 100 μm, LDTg 200 μm. (**F**) Quantification of CHAT positive neurons in the indicated cholinergic areas. ChAT-cKO mice do not express CHAT in the MHb, while striatum, PPTg and LDTg show comparable number of neurons positive for CHAT (percentage of the number of cells counted per section normalized to wt is shown in bars. MHb wt: 100.1 ± 3.676, number of sections=22; MHb ChAT-cKO: 0.3326 ± 0.1380, n=22; unpaired t-test p<0.0001; Striatum wt: 100.0 ± 7.233, n=22; Striatum ChAT-cKO: 92.27 ± 7.112, n=22; unpaired t-test p=0.45; PPTg wt: 100.1 ± 11.70, n=14; PPTg ChAT-cKO: 115.8 ± 10.78, n=14; unpaired t-test=0.33; LDTg wt: 100.0 ± 11.50, n=13; LDTg ChAT-cKO: 122.0 ± 16.86, n=13; unpaired t-test=0.29. The total number of neurons counted per brain area is shown above the bars; n=3 mice per genotype). (**G**) Sagittal brain sections immunostained for CHAT (red) and neurofilament (green). The axonal projections and general anatomy of the fr, IPN and surrounding tegmental areas are comparable between wt (left panels) and ChAT-cKO (right panels). ChAT-cKO mice lack CHAT-

*Figure 2 continued on next page*

*Figure 2 continued*

immunoreactivity in the MHb-fr-IPN projection but show similar CHAT expression as wt mice in other brain areas, such as the LDTg, third cranial nerve (3N), striatum (Str) and BF. Scale bar: 800 µm. BF: Basal forebrain; cKO: Conditional knockout; IPN: Interpeduncular nucleus; LDTg: Laterodorsal tegmental nucleus; MHb: Medial habenula.

## Cholinergic and glutamatergic inputs to IPN subnuclei

The major input to the IPN is glutamatergic and originates in the habenula, but there are several other glutamatergic afferents to the IPN. These include projections from the laterodorsal tegmentum (LDTg), raphe nuclei, locus coeruleus, periaqueductal grey (PAG) and nucleus of the diagonal band. Studies on the neurotransmitter content of these other inputs indicate that there are only three sources that send cholinergic and glutamatergic projections to the IPN: the MHb, the LDTg and the nucleus of the diagonal band. Importantly, in both the LDTg and the nucleus of the diagonal band, none of the glutamatergic cells overlap with cholinergic markers (*Henderson et al., 2010*; *Wang and Morales, 2009*). This implies that axonal terminals in the IPN that are colabeled with glutamatergic and cholinergic markers originate in the MHb. However, within the MHb there are two distinct populations: a cholinergic population in the ventral MHb that projects to the rostral (IPR), intermediate (IPI) and central (IPC) subnuclei of the IPN and a peptidergic (Substance P positive) population in the dorsal MHb that projects to the lateral part of the IPN (IPL)(*Figure 4A*). Both types of projections release glutamate. To quantify the degree of overlap of glutamatergic and cholinergic synapses in different nuclei of the IPN, we performed double immunostainings in wt and ChAT-cKO mice and calculated the Manders' colocalization coefficient which ranges from 0 for no colocalization to 1 for complete colocalization (*Manders et al., 1993*). These analyses indicated an extremely high colocalization of VGLUT1 and VACHT in the IPC of both wt (M1=0.80) and ChAT-CKO (M1=0.79), less overlap in the IPI (wt, M1=0.78, ChAT-CKO, M1=0.77) and IPR (wt, M1=0.68, ChAT-CKO, M1=0.69), and no colocalization in the IPL (wt, M1=0.005, ChAT-CKO, M1=0.01) (*Figure 4*). Taken

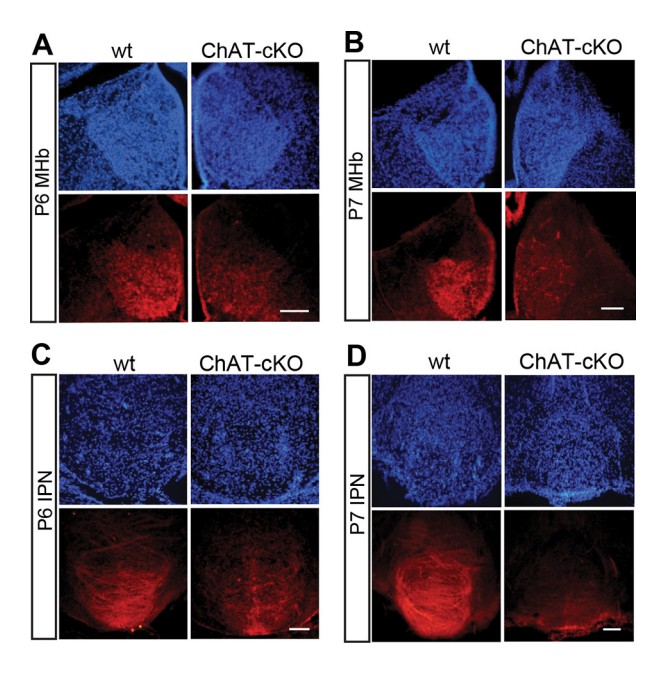

**Figure 3.** Conditional gene deletion of *Chat* driven by *Kiaa1107* Cre-recombinase line is specific for the MHb-IPN tract and occurs during the first postnatal week. (**A–D**) Immunostaining with CHAT (red) and counterstaining with DAPI (blue) shows progressive loss of CHAT signal in MHb neurons and their axonal terminals in the IPN at postnatal day 6 (P6) and postnatal day 7 (P7). IPN: Interpeduncular nucleus; MHb: Medial habenula.

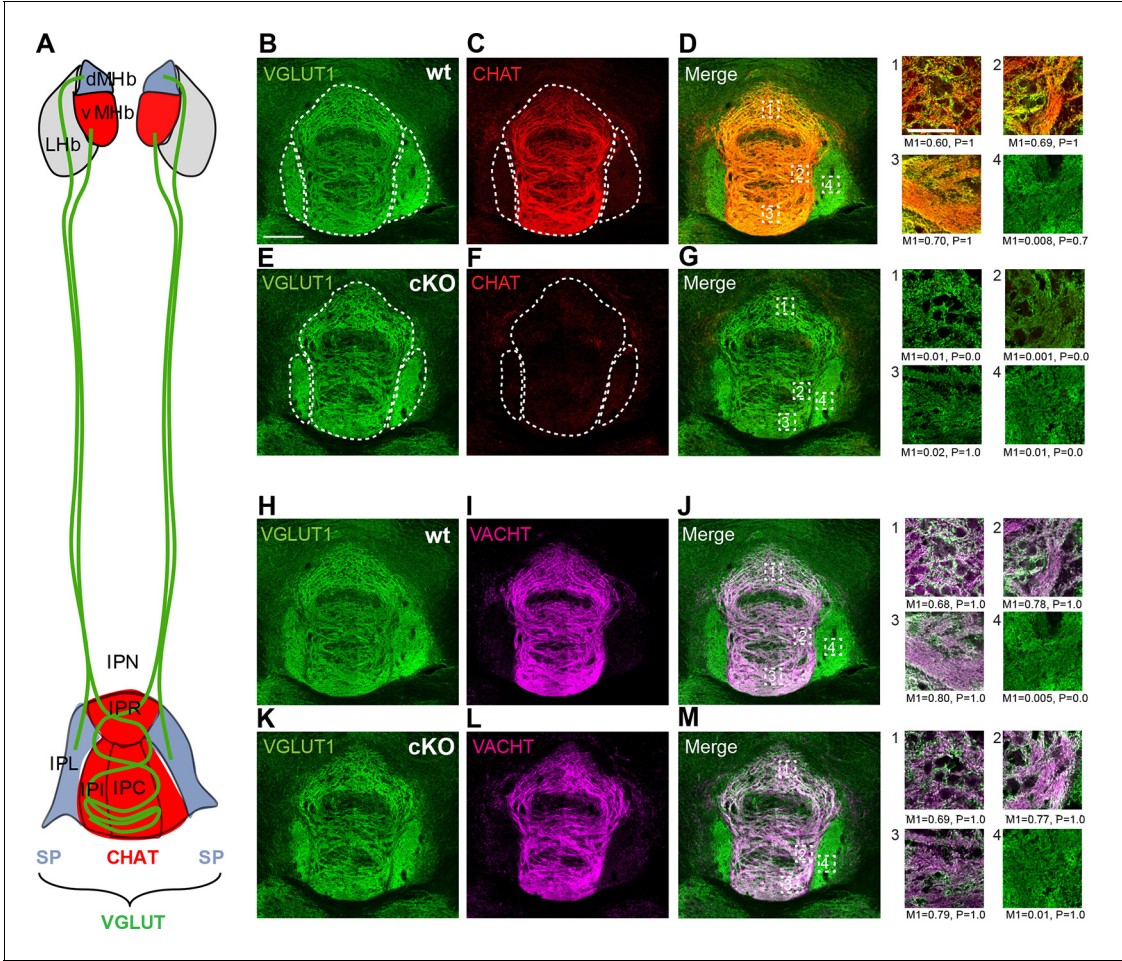

**Figure 4.** Distribution of cholinergic and glutamatergic axonal terminals in IPN subnuclei. (A) Schematic representation of the segregated projections of medial habenular neurons to IPN subnuclei. The dorsal part of the MHb (dMHb, blue) consists of peptidergic neurons (positive for substance P, SP) that project to the lateral part of the IPN (IPL, blue). The ventral part of the MHb (vMHb, red) contains cholinergic neurons (positive for CHAT, red) that project to the rostral (IPR), intermediate (IPI) and central (IPC) subnuclei of the IPN (red). Both types of projections co-release glutamate (VGLUT, green). (B–M) Double immunostaining and Manders' colocalization coefficient analyses of glutamatergic and cholinergic markers in different subnuclei of the IPN. M1: index of colocalization and P: significance of correlation were measured in the indicated dotted squares which are located: 1 in IPR, 2 in IPI, 3 in IPC and 4 in IPL. (B–G) In wt mice, VGLUT1 (green) and CHAT (red) show the strongest index of colocalization in the IPC and no colocalization in the IPL. CHAT immunoreactivity signal is absent in the IPN of ChAT-cKO mice. (H–M) Analyses of VGLUT1 (green) and VACHT (violet) show the highest colocalization index also in the IPC of both wt and ChAT-cKO, and no colocalization in the IPL. The M1 and the P values shown below panels 1–4 were calculated from one image for each IPN subnucleus. cKO: Conditional knockout; IPN: Interpeduncular nucleus; MHb: Medial habenula; wt: Wild type.

together with the published literature, our data demonstrate that the vast majority of cholinergic +glutamatergic input to the IPC originates in the ventral MHb.

## The vesicular transporters for acetylcholine and glutamate colocalize in the majority of synaptic vesicles of axonal terminals in the central IPN

We next wanted to evaluate whether the vesicular transporters for ACh and glutamate, VACHT and VGLUT1, which colocalize in the IPC by light microscopy (*Figure 4H–M*), actually label the same individual SV. Single immunogold electron microscopic experiments showed specific immunoreactivity for either VACHT or VGLUT1 at synaptic vesicular membranes in habenular presynaptic terminals (*Figure 5B–C*). Quantification analyses showed that both antibodies have comparable penetration and labeling immunoreactivities, including the number of VACHT and VGLUT1-labeled vesicles per synaptic terminal (*Figure 5D*), the labeling density (*Figure 5E*) and the percentage of vesicles labeled within a terminal (*Figure 5F*). In addition, the area of the presynaptic terminal and the length

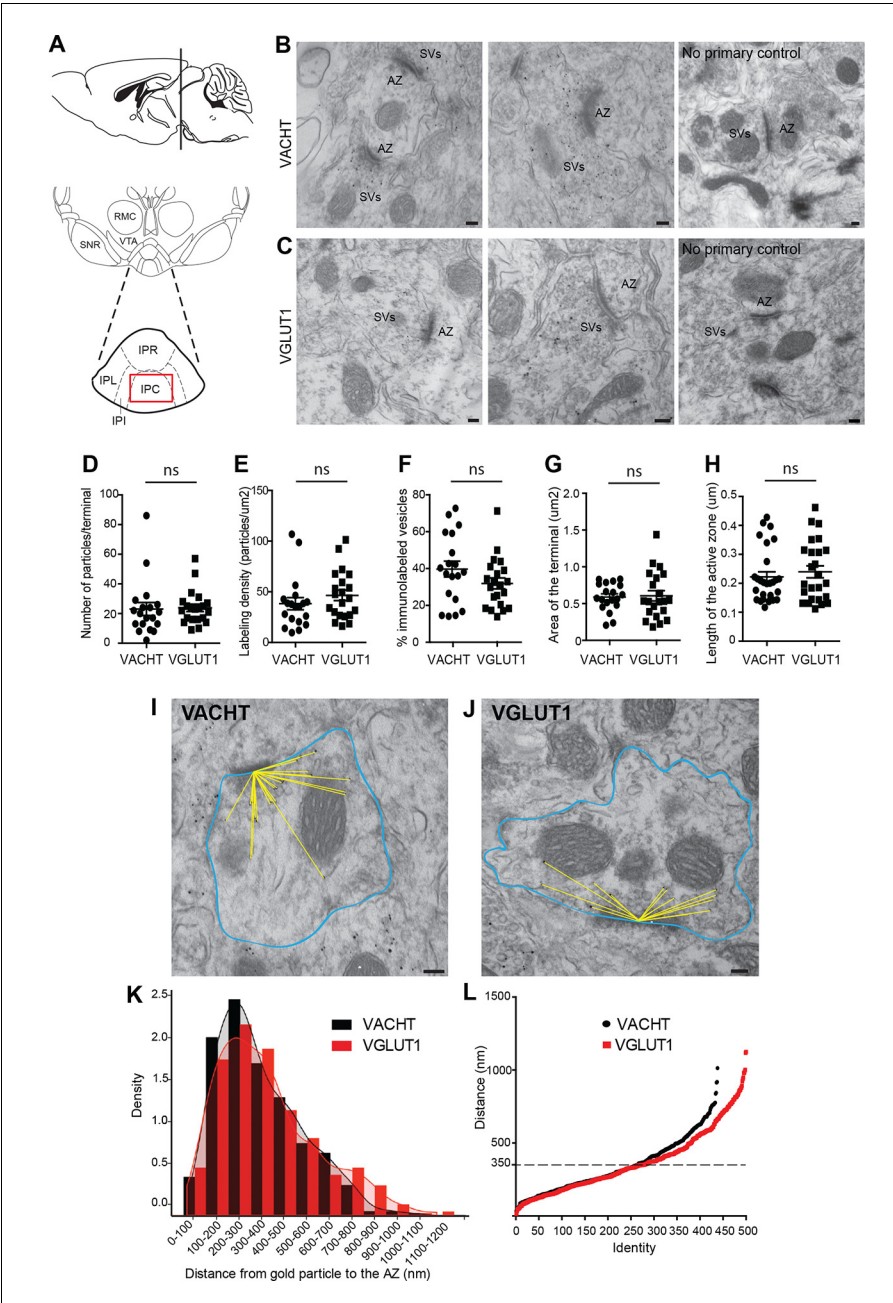

**Figure 5.** VACHT and VGLUT1 are located in synaptic vesicles of axonal terminals in the central IPN. (A) Scheme of the brain area dissected for Electron Microscopy (EM) analyses. The red square over the IPC indicates the region analyzed by EM. (B–C) Representative micrographs of single pre-embedding immunogold EM analyses of VACHT (B) and VGLUT1 (C) showing immunogold particles in synaptic vesicles (SVs) grouped together at presynaptic terminals identifiable by the dense active zone (AZ). The micrographs on the right column (no primary control) correspond to control experiments in which sections were treated following the same procedure while omitting the primary antibody. Scale bar: 100 nm. (D–H) Quantitative analysis of single pre-embedding immunogold electron micrographs. (D) The number of gold particles labeled either by VACHT or VGLUT1 antibodies per terminal is not statistical different between antibodies (VACHT=23.05 ± 4.37, n=19 terminals; VGLUT1= 23.86 ± 2.51, n=21 terminals; unpaired t-test p=0.87). (E) The labeling density (number of gold particles per terminal/area of the terminal) is not significantly different between VACHT and VGLUT1 positive terminals (VACHT= 38.31 ± 5.95, n=19; VGLUT1= 46.48 ± 5.175, n=21; unpaired t-test p=0.3). (F) No difference is observed in the percentage of VACHT or VGLUT1 immunolabeled vesicles (labeled vesicles/ total number of vesicles*100) per terminal (VACHT= 39.73 ± 4.30, n=19; VGLUT1= 31.91 ± 2.97, n=21; unpaired t-test p=0.13). (G) The area of the presynaptic terminals labeled with either VACHT or VGLUT1 gold particles is not significantly different (VACHT= 0.58 ± 0.04, n=19; VGLUT1= 0.61 ± 0.07, n=21; unpaired t-test p=0.84). (H) No difference is observed in the length of the active zone of presynaptic terminals labeled with either VACHT or VGLUT1 gold particles (VACHT= 0.22 ± 0.02, n=28; VGLUT1= 0.24 ± 0.02, n=25; unpaired t-test p=0.52). (I–L) For quantification of the distances of immunogold labeled SVs to the AZ, the area of each presynaptic terminal was delineated (blue) and

*Figure 5 continued on next page*

Figure 5 continued

the distance of each gold particle within a terminal to the center of the active zone (yellow lines) was measured for VACHT (I) and VGLUT1 (J). Scale bar: 100 nm. (K) Density plot of the distances of VACHT (black bars) and VGLUT1 (red bars) labeled particles to the active zone. (L) Scatter plot of the distances of VACHT and VGLUT1 gold particles to the active zone. Vesicles labeled by either VACHT or VGLUT1 located beyond 350 nm from the active zone are differentially distributed ($F_{(1,935)}$ = 4.634; $p < 0.05$; one-way ANOVA).

of the active zone labeled by each antibody were similar in both groups (*Figure 5G, H*) indicating that the localization of VACHT and VGLUT1 in synaptic terminals is comparable. Given that the extent of labeling of SVs at habenular synapses is similar and high using these antibodies, these reagents can be used to study in detail the distribution, density and colocalization of VACHT and VGLUT1 transporters in SVs.

To determine whether they were differentially distributed across pools of synaptic vesicles, we measured the distance between each VACHT and VGLUT1-labeled vesicle and the center of the closest active zone (*Figure 5I, J*). There were no significant differences in the density of VACHT and VGLUT-1-labeled vesicles within 350 nm from the active zone (*Figure 5K–L*). However the scatter plot distribution showed that VACHT and VGLUT1-labeled vesicles located farther than 350 nm from the active zone were differentially distributed ($F(1,936)$ = 4.954; $p < 0.05$; one-way ANOVA) (*Figure 5L*). It has been proposed that SVs might be organized in three functionally distinct pools at increasing distances from the active zone: the readily releasable pool (RRP 0–60 nm), the recycling pool (60–200 nm), and the reserve pool (beyond 200 nm) (*Rizzoli and Betz, 2005*); however it is still debated whether the recycling and reserve pools are intermingled (*Marra et al., 2012*; *Rizzoli, 2014*). Our results show similar distribution of vesicles containing ACh and glutamate within the two closer SV pools. However the reserve pool appears to contain more glutamatergic than cholinergic SVs.

Finally, to determine whether VGLUT1 and VACHT are present in the same synapses and synaptic vesicles, we performed double post-embedding immunogold electron microscopic analyses (*Figure 6A, B*). We observed colabeling of VACHT-6 nm and VGLUT1-12 nm gold particles in 72 of 90 labeled synaptic terminals, 8 terminals with only VAChT-6 nm particles and 10 terminals with only VGLUT1-12 nm particles indicating that both transporters are present in the vast majority (80%) of habenular terminals in the central IPN (*Figure 6C*). To assess whether VACHT and VGLUT1 are present in the same SV, we measured the distance from each VGLUT1-gold particle to the nearest VACHT-gold particle (*Figure 6D*) and adopted the criterion described by (*Stensrud et al., 2013*) stating that 90 nm is the maximal distance between gold particles which colocalize in the same vesicle (*Figure 6E*). This distance has been calculated based on the diameter of a synaptic vesicle of approximately 30 nm (*Gundersen et al., 1998*), which we consistently found to be 35.18 ± 0.43 nm (n=203) in habenular terminals, and based on the distance from an antigen epitope to the center of a secondary antibody conjugated with 10 nm gold particles which spans up to 30 nm (*Bergersen et al., 2012*). Hence, two gold particles located within a distance of 90 nm could potentially label the same vesicle (*Stensrud et al., 2013*). Considering this maximal theoretical distance of 90 nm, 54.35% of VGLUT1-gold particles (n=75/138) could be located in the same SV as VACHT (*Figure 6D*). However given that SVs are closely packed, this may be an overestimate because adjacent particles could also be labeling transporters in neighboring SVs. As control analyses, we quantified the distance from each VGLUT1-gold particle to the closest VGLUT1-particle (*Figure 6F*), and the distance from each VACHT to the closest VACHT-gold particle (*Figure 6G*). We found that 35.95% of VGLUT1 particles (n=32/89), and 61.08% of VACHT particles (n=113/185) are within 90 nm of each other and could potentially label the same synaptic vesicle. These ultrastructural analyses indicate that ACh and glutamate vesicular transporters are coexpressed in 80% of the axonal terminals in the IPC, and that within each presynaptic terminal a significant fraction of SVs colabel with VACHT and VGLUT immunogold particles, supporting the conclusion that ACh and glutamate can be cotransported into the same synaptic vesicles in habenular cholinergic presynaptic terminals

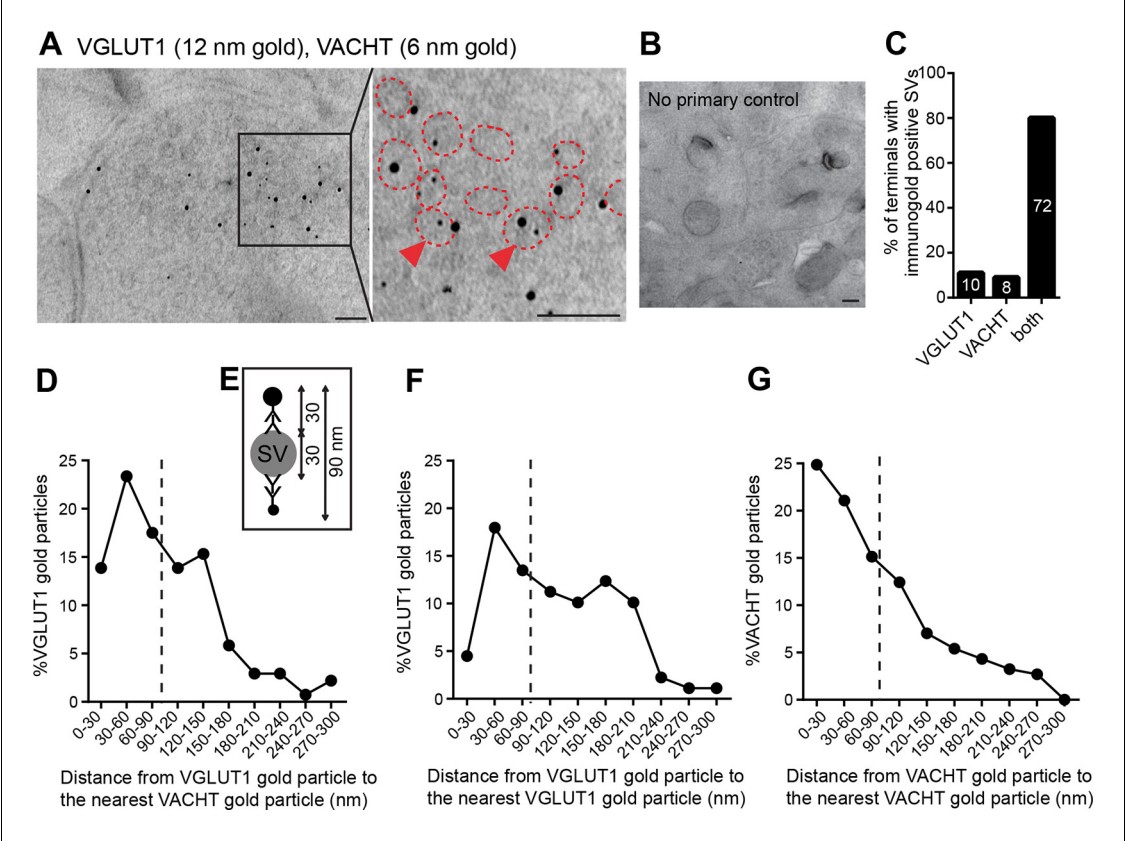

**Figure 6.** VACHT and VGLUT1 colocalize in the same synaptic vesicles in axonal terminals in the central IPN. (**A,B**) Double post-embedding immunogold labeling of VACHT (6 nm gold particles) and VGLUT1 (12 nm gold particles) shows colocalization of both transporters in the same synaptic vesicles. The right panel is a higher magnification of the boxed area. Synaptic vesicles are outlined in red. Arrowheads indicate synaptic vesicles that contain both transporters. Scale bar: 100 nm. In control experiments sections were treated following the same procedure while omitting the primary antibodies. Scale bar: 100 nm. (**C**) Percentage of axonal terminals that contain VGLUT1 or/and VACHT immunolabeled SVs. The number of counted terminals is shown inside each bar of the graph: from 109 visualized terminals, 10 terminals contain only VGLUT1 positive SVs, 8 terminals contain only VACHT positive SVs, 72 terminals contain SVs positive for VGLUT1 and VACHT, and 19 terminals showed no immunogold particles. (**D**) Frequency histogram showing the percentage of VGLUT1 particles located at the indicated distances (30 nm bins) from the nearest VACHT gold particle. (**E**) Scheme of an SV labeled with two primary antibodies and secondary antibodies conjugated to a gold particle. According to the indicated size of an SV and conjugated antibodies, the maximum distance between two immunogold particles to potentially label the same synaptic vesicle is 90 nm. (**D,F,G**) Frequency histograms of the distance distributions between immunogold particles showing the percentage of particles located at the indicated distances (30 nm bins) for VGLUT1 to VACHT (**D**), for VGLUT1 to VGLUT1 (**F**) and for VACHT to VACHT (**G**). Particles located closer than 90 nm apart (on the left side of the dashed line) potentially label the same vesicle. SV: Synaptic vesicle

## Local elimination of ACh in medial habenular neurons reduces glutamate corelease

Given our finding that glutamate and ACh can be cotransported into SVs (*Figure 6*), and that release of glutamate and ACh activate postsynaptic receptors via wired and volume transmission, respectively (*Ren et al., 2011*), we were interested in the detailed mechanisms through which the lack of ACh could affect cholinergic and glutamatergic transmission. nAChRs are highly enriched in three compartments of the MHb-IPN tract (*Perry et al., 2002*; *Zoli et al., 1995*): the soma of MHb neurons, presynaptic terminals of MHb neurons in the IPN, and postsynaptic IPN neurons (*Covernton and Lester, 2002*; *Frahm et al., 2011*; *Girod et al., 2000*; *Ren et al., 2011*). In contrast to extrasynaptic nAChRs that are activated by ambient levels of ACh by volume transmission, glutamate receptors are present on the postsynaptic IPN membrane and they rapidly (milliseconds) respond to fast stimulation in a wired transmission mode (*Ren et al., 2011*). To distinguish cholinergic and glutamatergic currents in wt and ChAT-cKO mice we performed electrophysiological whole-cell recordings using different pharmacological conditions.

We first measured evoked excitatory postsynaptic currents (eEPSCs) by local puff application of nicotine (100 ms, 100 µM). In MHb neurons, nicotine elicits large eEPSCs (>2000 pA) that are blocked by nAChR blockers (*Frahm et al., 2011*) and are thus exclusively cholinergic. As expected, eEPSCs did not differ between wt and ChAT-cKO (wt: 2265.3 ± 210.4 pA, n=18; cKO: 2467 ± 175.2 pA, n=18; p=0.465) (*Figure 7B,C*), since conditional deletion of *Chat* in the MHb would not directly affect expression of any cholinergic genes in the *stria medullaris*, which gives rise to the presynaptic cholinergic input to the MHb (*Contestabile and Fonnum, 1983*; *Gottesfeld and Jacobowitz, 1979*). We next asked whether IPC neurons, which receive cholinergic afferents mostly from the MHb (*Figures 4–6*), respond differently in wt and ChAT-cKO. It has been shown that ACh (1 mM) evokes slow (seconds) inward currents from IPN neurons that are reversibly abolished by nAChR blockers (*Ren et al., 2011*), and that optogenetic tetanic stimulation generates slow cholinergic inward currents as well as many fast glutamatergic inward currents at high-frequency, indicating that eEPSCs are mainly cholinergic but have a fast glutamatergic component. Pressure application of nicotine (100 µM) evoked medium size inward currents from IPN neurons that had similar amplitudes in wt and ChAT-cKO mice IPN: wt: 254.3 ± 45.55 pA, n=52; cKO: 304.8 ± 51.03 pA, n=53; p=0.475) (*Figure 7D,E*), demonstrating that elimination of ACh in habenular neurons does not affect nicotine-evoked responses of postsynaptic nAChRs in MHb and IPN neurons.

We next recorded IPN neurons which exhibit fast spontaneous synaptic currents, mediated by AMPA-type glutamate receptors and $GABA_A$-type receptors (*Ren et al., 2011*). To isolate miniature excitatory postsynaptic potentials (mEPSCs), we performed the recordings in the presence of $GABA_A$ receptor blockers and tetrodotoxin (TTX) (*Figure 7 F–K*). TTX blocks action potential formation and its propagation. Thus, mEPSCs events reflect the probabilistic release of single vesicles into the synapse and their measurement can be used for the quantification of release probability and vesicular glutamate content (*Pinheiro and Mulle, 2008*). mEPSC were blocked by glutamate receptor inhibitors NBQX and D-AP5 (*Figure 7H*) indicating that mEPSCs in IPC neurons are glutamatergic, not cholinergic. Because IPN neurons exhibit high frequency mEPSCs, we only analyzed fast mEPSCs that exhibited an average amplitude >5 pA. This analysis revealed that the average amplitude of mEPSC in IPC neurons of wt mice is 18.2 ± 1.6 pA; whereas the average amplitude in ChAT-cKO neurons is 12.4 ± 0.9 pA (p=0.005; wt: n=25; ChAT-cKO: n=24) (*Figure 7J*). This corresponds to a 32% reduction in the amplitudes of glutamatergic mEPSC in ChAT-cKO animals, which might reflect a decrease of the vesicular content of glutamate or differences in the number or function of postsynaptic glutamate receptors (*Liu et al., 1999*; *Watt et al., 2000*). Quantitative analyses of the frequency of mEPSC showed no differences between IPC neurons of wt (5.3 ± 1.2 Hz for wt) and ChAT-cKO mice (3.9 ± 0.8 Hz) (*Figure 7K*). The fact that glutamatergic mEPSCs have significantly reduced amplitudes (but similar frequencies) in central IPN neurons of ChAT-cKO mice, and that nicotine-evoked cholinergic eEPSCs remain unchanged in these mice, suggest that elimination of habenular ACh decreases the vesicular content of glutamate but does not impact on the excitability of postsynaptic nAChRs.

## Presynaptic facilitation of glutamate release is absent in ChAT-cKO mice

It has been reported that presynaptic nAChRs facilitate glutamate release in a variety of synapses, including MHb-IPN terminals (*Girod et al., 2000*; *Girod and Role, 2001*; *McGehee et al., 1995*). To test whether local release of ACh could influence presynaptic facilitation by activation of nAChRs, we recorded mEPSC upon bath application of nicotine or ACh. Superfusion with nicotine (*Figure 8B*) or ACh (*Figure 8C*) did not change the amplitudes of mEPSCs in either wt, or increased the amplitudes of ChAT-cKO to wt levels, consistent with the hypothesis that the decrease of glutamate vesicular content in presynaptic terminals depends on the presence of intracellular ACh. Importantly, we observed a significant increase in mEPSC frequency in wt, but not in ChAT-cKO mice upon superfusion with nicotine (*Figure 8D*) and ACh (*Figure 8E*). This indicates that in the absence of network activity eliminated by TTX, nicotine or ACh activation of nAChRs at presynaptic terminals elicits enhanced discharge of glutamatergic mEPSCs in IPC neurons of wt mice. In contrast there is no presynaptic facilitation in ChAT-cKO neurons, suggesting that presynaptic nAChRs are downregulated in the absence of ACh released by habenular terminals. We next analyzed holding currents as a measure of the excitability of postsynaptic IPN neurons. The values are very similar between genotypes and upon perfusion with nAChR agonists (*Figure 8F*), indicating that postsynaptic

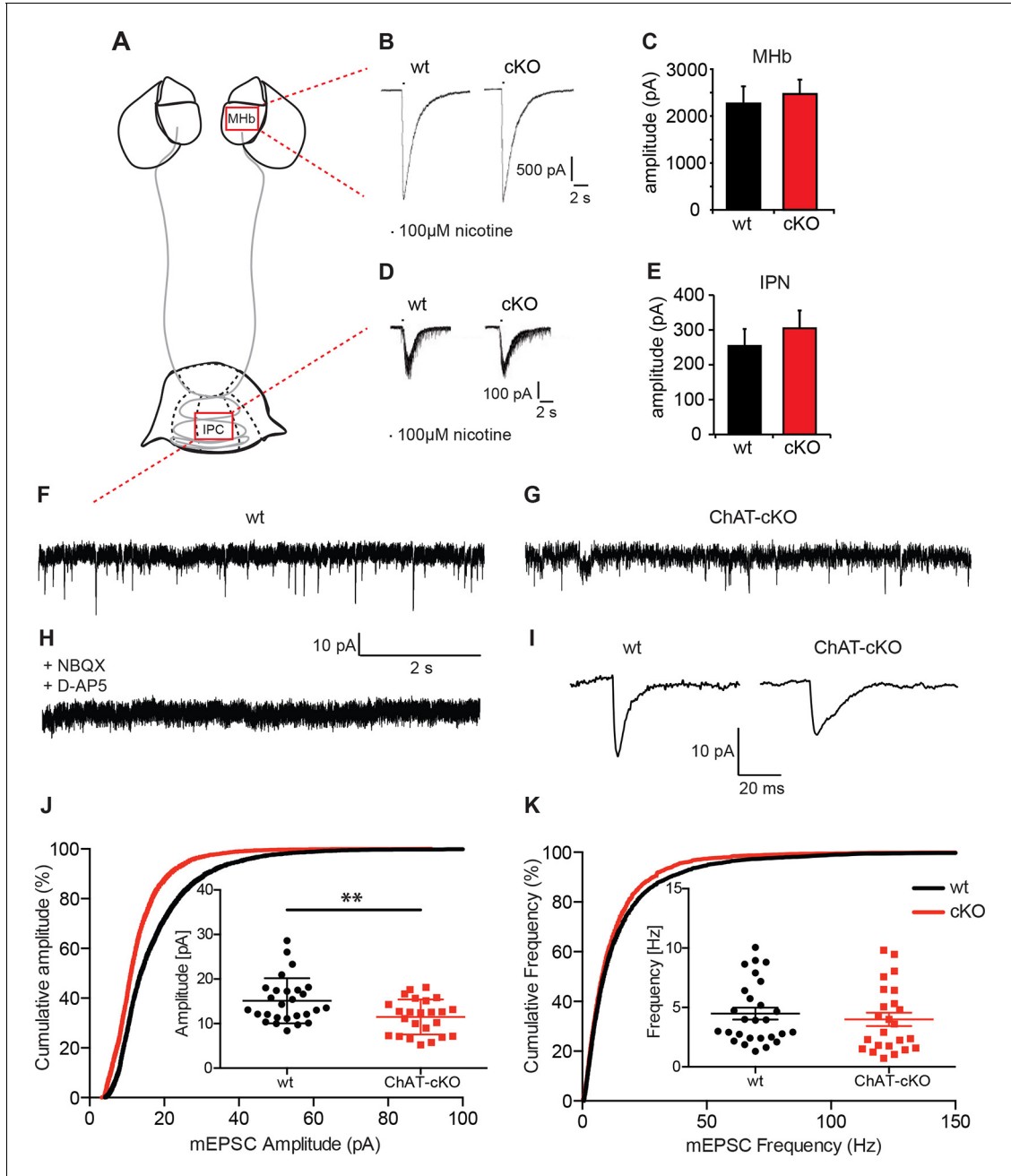

**Figure 7.** Cholinergic eEPSC are unchanged whereas glutamatergic mEPSC are smaller in central IPN neurons of ChAT-cKO mice. (A) Schematic representation of the habenula and IPN subnuclei. Red squares over the MHb and IPC indicate the areas where the corresponding electrophysiological recordings were performed. (B–E) Cholinergic representative evoked excitatory postsynaptic currents (eEPSC) of MHb (B) and IPC (D) neurons by local puff application of 100 µM nicotine. Cholinergic, current amplitudes were comparable between wt and ChAT-cKO. (C) n= 18 per genotype, unpaired t-test p=0.465. (E) n= 52 (wt), n= 53 (ChAT-cKO), unpaired t-test p=0.475. (F–I) Representative recordings of glutamatergic miniature excitatory postsynaptic currents (mEPSC) in IPC neurons in wt (F), ChAT-cKO (G), and in wt neurons after addition of the AMPA-R and NMDA-R blockers NBQX and D-AP5 (H). (I) Representative average glutamatergic mEPSC in wt and ChAT-cKO. (J) Cumulative probability plots of the glutamatergic mEPSC amplitudes show a left shift indicating that ChAT-cKO neurons (red line) have significantly smaller amplitudes than wt neurons (black line). Insets represent the average amplitude of mEPSCs per neuron recorded during 1 min (n=24 for wt and n=24 for cKO, unpaired t-test: **p<0.01). (K) Cumulative probability plots of the mEPSC frequency show similar curves for ChAT-cKO neurons (red line) and wt neurons (black line). Insets show no significant differences in the average frequencies per recorded neuron (n=27 for wt and n=24 for cKO). cKO: Conditional knockout; IPN: Interpeduncular nucleus.

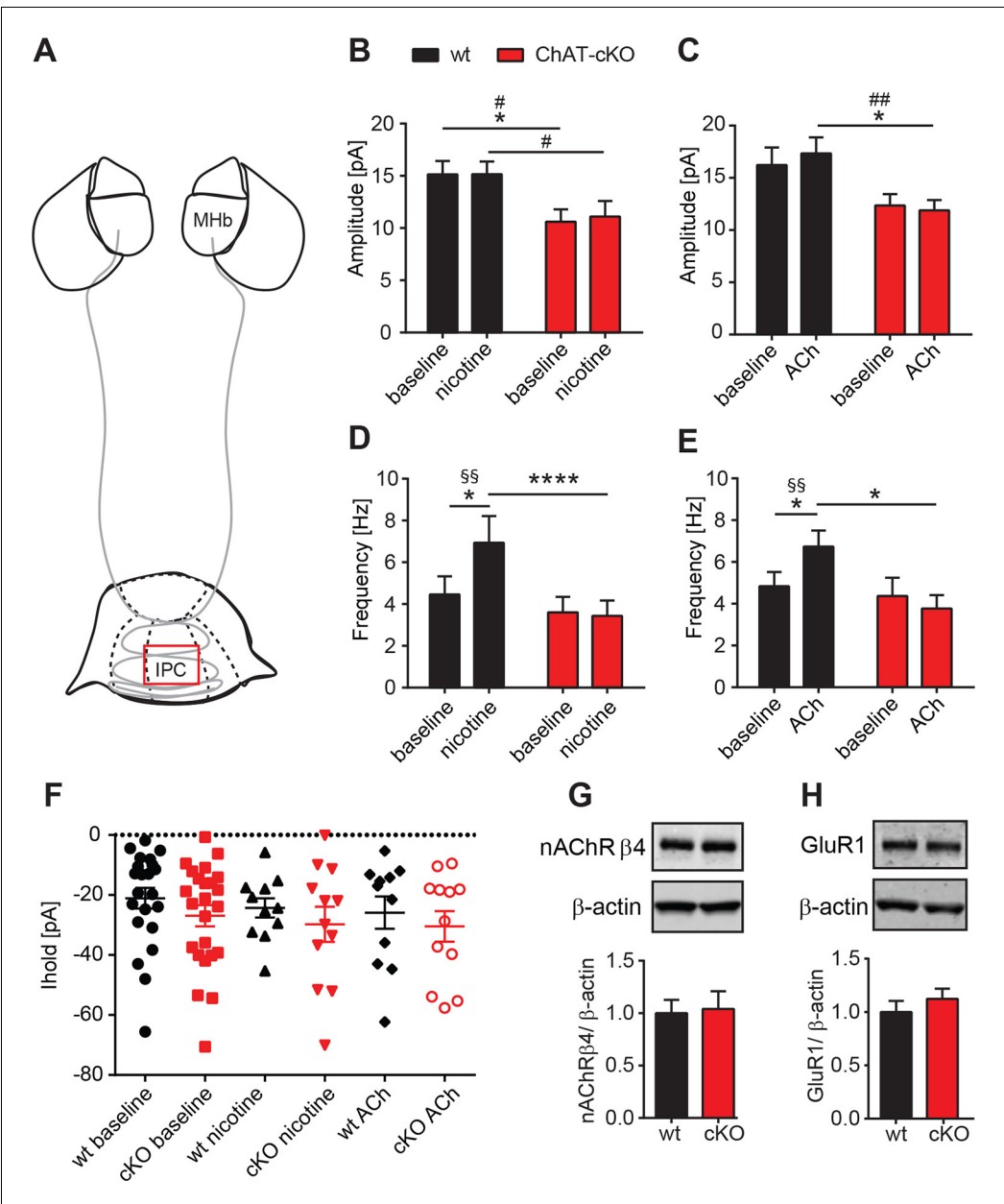

**Figure 8.** Presynaptic facilitation of glutamate release is absent in ChAT-cKO mice. (A) Schematic representation of the habenula and IPN subnuclei. Red square over the IPC indicates the area of electrophysiological recordings. (B,C) Superfusion with 250 nM nicotine (B) or 100 µM ACh (C) did not change the glutamatergic mEPSC amplitude within each group (n=22–24 per genotype, n=11–12 per condition). ChAT-cKO neurons show significantly smaller amplitudes than wt neurons in either condition (repeated measures two-way-ANOVA: p<0.05 for genotype (for nicotine and ACh), Bonferroni's posttest: *p<0.05 and unpaired t-test: # p<0.05, ## p<0.01). (D,E) Bath application of 250 nM nicotine (D) or 100 µM ACh (E) significantly increased the glutamatergic mEPSC frequency in wt, but not in ChAT-cKO (n=22–24 per genotype and n=11–12 per condition, repeated measures two-way-ANOVA: p<0.01 for treatment and p<0.001 for interaction in F and p<0.05 for interaction in G; Bonferroni's posttest: *p<0.05, ****p<0.0001 and paired t-test: §§ p<0.01). (F) The holding current is similar in wt and cKO and is not altered by bath application of 250 nM nicotine or 100 µM ACh (n=22–24 per genotype, n=11–12 per condition). (G,H) Western blot analyses and quantification of β4 and GluR1 levels in IPN membrane extracts reveal no significant differences between wt and ChAT-cKO mice (n= 3 per genotype).

nAChRs in IPN neurons are not affected by elimination of endogenous ACh. Western blot analyses of nAChRs and GluR1 in IPN extracts did not show significant differences between wt and ChAT-cKO (*Figure 8G–H*). However this assay cannot distinguish between presynaptic, extrasynaptic and postsynaptic receptors, and whether these are at the membrane or not. Given the nAChR-mediated

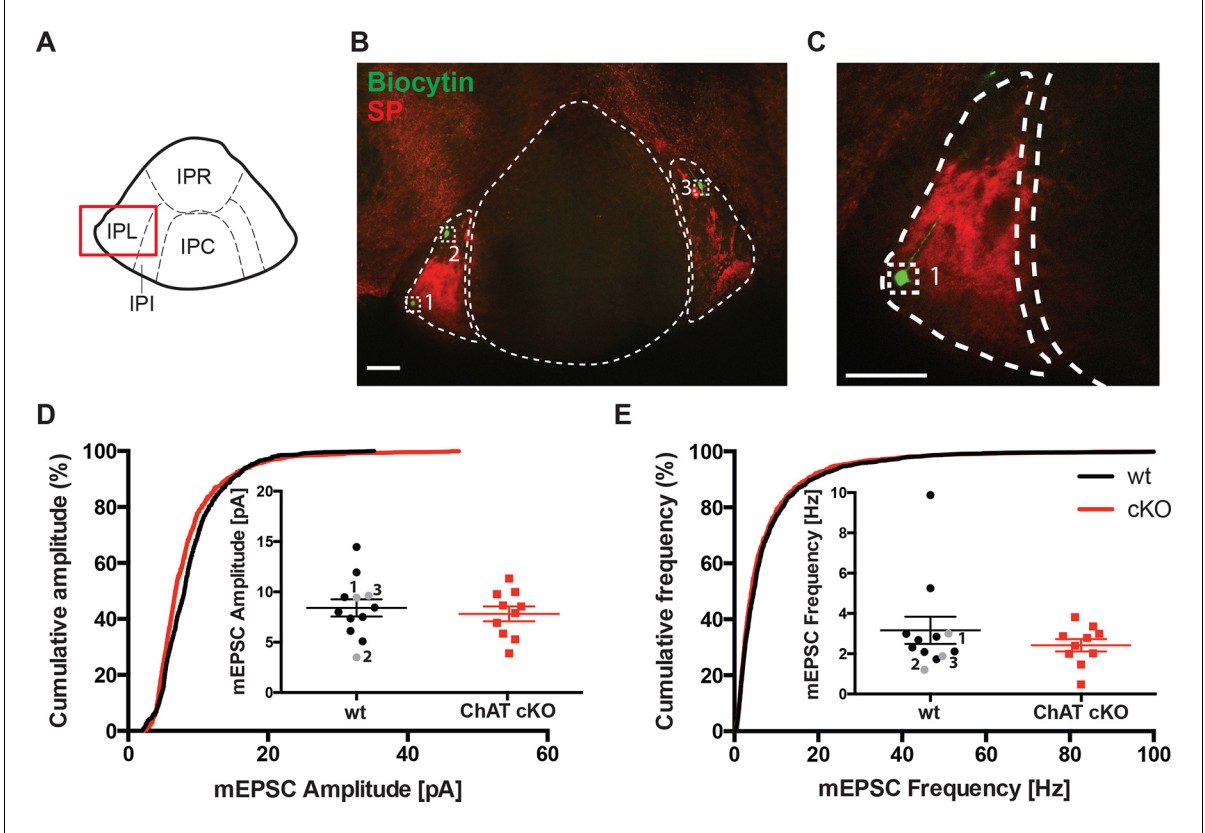

**Figure 9.** Glutamatergic mEPSC are unchanged in the lateral part of the IPN. (A) Schematic representation of IPN subnuclei. Red square over the IPL indicates the area of electrophysiological recordings. The IPL does not receive cholinergic afferents from the ventral part of MHb, but mostly substance P afferents from the dorsal part of MHb. (B) Representative image of three recorded neurons filled with Biocytin (green) in the substance P (SP; red) positive IPL. Scale bar: 100 μm. (C) Higher magnification of neuron 1 in the IPL. Scale bar: 100 μm. (D) Cumulative and average amplitude of glutamatergic mEPSC recordings in the IPL. mEPSC amplitudes were not significantly different between wt (n=12) and ChAT-cKO (n=10; unpaired t-test p=0.556). Marked in grey are the recorded cells 1, 2 and 3 shown in B and C. (E) Cumulative and average frequency of glutamatergic mEPSC recordings in the IPL. There were no significant differences in mEPSC frequency between wt (n=12) and ChAT-cKO (n=10; unpaired t-test p=0.364). Marked in grey are the recorded cells 1, 2 and 3 shown in B and C. cKO: Conditional knockout; IPN: Interpeduncular nucleus; IPL: Lateral interpeduncular nucleus; mEPSC: miniature excitatory postsynaptic currents; MHb: Medial habenula; wt: Wild type.

presynaptic effect we have measured, it is possible that ACh might be required for the trafficking of nAChRs to the membrane without affecting the total number of nAChRs in the IPN. Taken together, these data and *Figure 7* show that the function of presynaptic but not postsynaptic nAChR receptors is reduced in the IPC of ChAT-cKO, and that ACh is necessary for presynaptic nAChR-mediated facilitation and for increasing the vesicular glutamate content.

## Glutamatergic non-cholinergic mEPSC are unchanged in the lateral part of the IPN

To confirm that glutamatergic inputs to the IPN that are non-cholinergic are not affected by selective elimination of ACh in habenular neurons, we performed electrophysiological recordings in the IPL (*Figure 9A*), which receives non-cholinergic glutamatergic inputs from substance P positive neurons in the ventral MHb (*Figure 4*) (*Antolin-Fontes et al., 2015*). Cumulative and average amplitude of glutamatergic mEPSC recordings in IPL neurons identified by biocytin and SP immunostaings (*Figure 9B,C*) showed that mEPSC amplitudes and frequencies were not significantly different between wt and ChAT-cKO (*Figure 9D,E*). It is interesting to note that in wt mice mEPSC were smaller in the IPL with respect to the IPC and similar to the amplitudes detected in the IPC of ChAT-cKO mice (*Figure 7J* and *9D*). Altogether these results support the additional conclusion that

glutamatergic mEPSC are indeed of higher amplitude if ACh is present, suggesting that cotransmission with ACh might be a general mechanism to enhance glutamatergic transmission.

## ACh cotransport into synaptic vesicles increases glutamate uptake

To determine if indeed ACh promotes the uptake of glutamate into SVs we assessed the levels of vesicular transporters in ChAT-cKO and wt mice by WB and coimmunoisolations, and performed glutamate uptake experiments. Western blot analyses of isolated SVs revealed no differences in the protein levels of Synaptophysin (Syp), VACHT, VGLUT1, and VGLUT2 between wt and ChAT-cKO mice (*Figure 10A*: lanes 1 and 6 and *Figure 10B*) indicating that loss of CHAT did not alter the expression of synaptic vesicle proteins. Immunoisolations of SVs from wt and ChAT-cKO mice using polyclonal antibodies directed against Syp, VGLUT1, VGLUT2, and VACHT (*Figure 10A*) and immunoisolations with IgG antibodies as background control revealed that VGLUT1-isolated SVs were positive for VACHT and that VACHT-isolated SVs were positive for VGLUT1 (yellow squares, *Figure 10A*). In all cases, the immune-detected signals were comparable in wt and ChAT-cKO vesicles. Additionally, a portion of VGLUT2-specific immunoisolates showed coexistence of VACHT in wt and ChAT-cKO vesicles (blue squares *Figure 10A*). Immunostaining of brain sections containing the IPN confirmed the expression of both VGLUT1 and VGLUT2 glutamate transporters in the IPN (*Figure 10C*). Coexpression of both transporters in habenular terminals in the IPN has been reported in the rat (*Aizawa et al., 2012*). Together, these results indicate the coexistence of VGLUT1/ VGLUT2, VGLUT1/VACHT and VGLUT2/VACHT in synaptic vesicles of the IPN and show that ChAT-cKO mice express normal levels of these vesicular transporters.

We next analyzed glutamate uptake into SVs isolated from rat IPN. Given that VACHT and VGLUTs are coexpressed in cholinergic SVs (*Figures 4–6* and *10A–C*) and that the absence of ACh decreases mEPSC amplitudes (*Figure 7*), we hypothesized that the addition of ACh would increase the uptake of glutamate into SVs. Vesicular transporters use the energy generated by the v-ATPase that pumps protons into the SV to generate an acidic ($\Delta pH$) and positively charged ($\Delta\psi$) electrochemical gradient ($\Delta\mu H^+$) (*Figure 10D*). During each transport cycle, two protons will be replaced by one ACh thereby decreasing $\Delta pH$ without affecting the positive outwardly directed $\Delta\psi$, a situation promoting VGLUT activity (*Figure 10D*) (*Hnasko and Edwards, 2012*). The addition of $NH_4^+$ increases vesicular glutamate uptake by increasing the intravesicular pH therefore lowering $\Delta pH$ without affecting $\Delta\psi$ (*Preobraschenski et al., 2014*), indicating that ACh might work in the same direction. While recent evidence indicates that vesicular synergy is operated by glutamate acting as a buffering anion (*Gras et al., 2008*), the reciprocal synergy has not been reported. To test this possibility, we purified SVs from IPN samples of rats and performed uptake assays. Uptake of [$^3$H]-glutamate was measured in the presence of neostigmine, a specific inhibitor of the AChE. Values were corrected for nonspecific uptake in the presence of FCCP (Carbonyl cyanide 4-(trifluoromethoxy) phenylhydrazone; inhibitor of vesicular transporters). Under these conditions, the accumulation of [$^3$H]-glutamate in IPN vesicles increased and reached a plateau of 38 ± 3 pmol/ mg of protein at 10 min (*Figure 10E*). In the presence of ACh, [$^3$H]-glutamate accumulation increased to 51 ± 15 pmol of glutamate/mg of protein (*Figure 10E*). Thus, addition of ACh increased vesicular glutamate uptake by 35.8 ± 13.6% relative to control vesicles. Addition of the positively charged $NH_4^+$ also led to an increase of glutamate uptake (140.7 ± 13 pmol). The increase by $NH_4^+$ was much more pronounced than with ACh possibly because $NH_4^+$ addresses all glutamatergic SVs present in the IPN preparation, while ACh can only increase glutamate uptake in those SVs equipped with VACHT and VGLUT. Altogether, our results show that ACh synergizes vesicular filling of glutamate, providing additional functional evidence for the coexistence of VACHT and glutamate vesicular transporters in a subset of IPN vesicles.

## ChAT-cKO mice do not display nicotine-mediated reward and withdrawal behaviors

Given that local elimination of ACh in habenular neurons reduces glutamate corelease (*Figure 7*) and impairs nAChR presynaptic facilitation (*Figure 8*), and given the role for nAChRs in the MHb/ IPN in nicotine craving and intake (*Fowler et al., 2011*; *Frahm et al., 2011*; *Salas et al., 2009*), we examined the behavioral responses of ChAT-cKO mice to nicotine. Psychomotor responses after acute nicotine challenge reflect the sensitivity of an individual to nicotine (*Clarke and Kumar, 1983*).

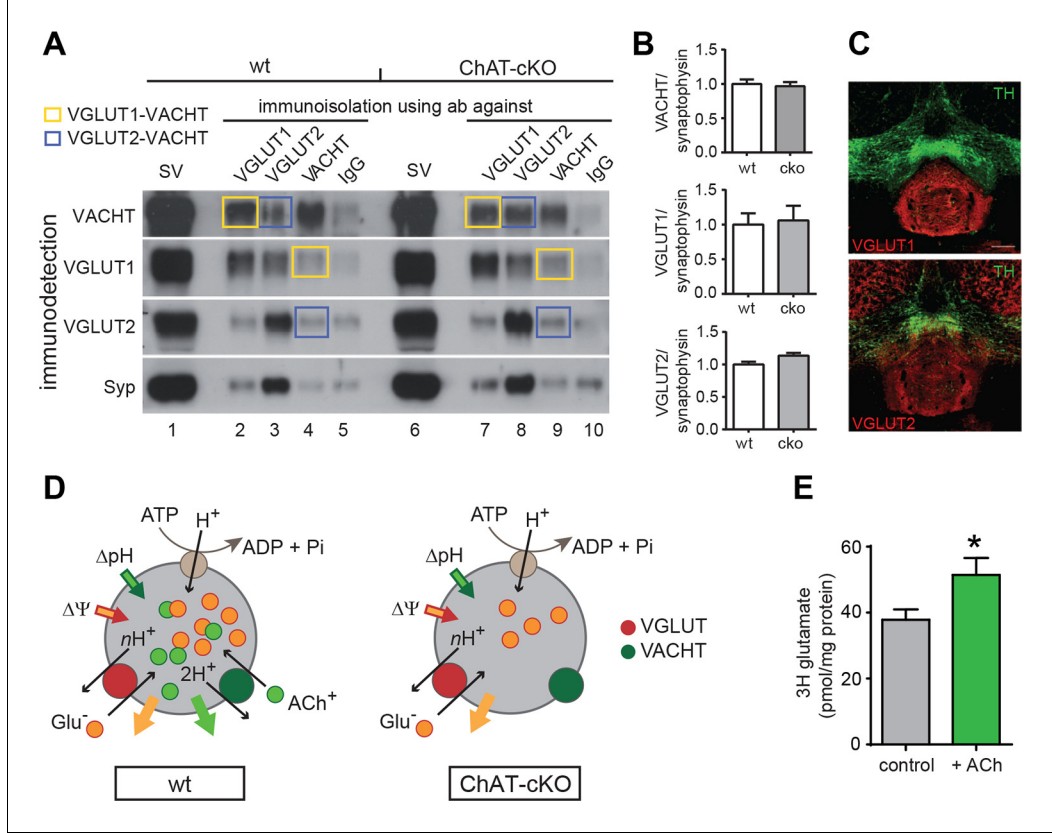

**Figure 10.** ACh enhances glutamate uptake into synaptic vesicles coexpressing VACHT and VGLUT1/2. (**A**) Western blot analyses of synaptic vesicles prepared from the IPN of wt and ChAT-cKO mice using VACHT, VGLUT1, VGLUT2, and Synaptophysin (Syp) antibodies indicated on the left. SVs from the starting material (shown in the first row for each genotype) were immunoisolated with the polyclonal antibodies indicated above, or without primary antibody (IgG) and subsequently immunodetected with the monoclonal antibodies indicated on the left. SVs immunoisolated with VGLUT1 contain VACHT and SVs immunoisolated with VACHT contain VGLUT1 and are indicated by yellow squares. Blue squares indicate VGLUT2/VACHT co-immunoisolates. (**B**) Quantification of the western blot signal of vesicular proteins in the starting fraction of the immunoisolation shows no significant differences between wt and ChAT-cKO. (**C**) Immunostaining of mouse brain sections containing the IPN and adjacent dopaminergic VTA region (TH positive in green) shows expression of VGLUT1 (red, upper panel) and VGLUT2 (red, lower panel). (**D**) Simplified illustration of vesicular synergy: synaptic vesicles are acidified and positively charged by a V-type $H^+$-ATPase (brown circle) that pumps protons into the SV generating a chemical gradient ($\Delta pH$, inward green arrow) and an electrical gradient ($\Delta \Psi$, inward red arrow). The activities of vesicular transporters depend to differing extents on $\Delta pH$ and $\Delta \Psi$ due to the charge on the neurotransmitter and the stoichiometry of coupling to $H^+$. VACHT preferentially uses the $\Delta pH$ chemical gradient while VGLUT relies more on $\Delta \Psi$ component. During each transport cycle, VACHT exchanges two protons ($H^+$) for one ACh molecule ($ACh^+$ positively charged) thereby dissipating $\Delta pH$ (outward green arrow) twice faster than $\Delta \Psi$, a situation promoting VGLUT activity. VGLUT activity exchanges one glutamate molecule ($Glu^-$ negatively charged) by $nH^+$ and causes the opposite imbalance dissipating more $\Delta \Psi$ than $\Delta pH$ (outward orange arrow). The transport activity of VGLUT thereby compensates for the bioenergetic imbalance produced by VACHT activity and vice versa. This reciprocal vesicular synergy between transporters allows a maximal accumulation of ACh and glutamate under a constant $\Delta \Psi$ and $\Delta pH$ gradient (see review by *Hnasko et al., 2012*). In the absence of $ACh^+$ as it occurs in synaptic terminals of ChAT-cKO mice, there is no exchange of $ACh^+/2H^+$ and synaptic vesicles progressively become less positively charged and the plateau level of $Glu^-$ uptake is reduced. Adapted from (*Gras et al., 2008*). (**E**) Glutamate uptake into SV prepared from rat IPN. In the presence of 10 mM ACh iodide (10 mM K-iodide was used for osmotic compensation in control condition), vesicles accumulated significantly more [$^3$H]glutamate (mean ± S.E.M. of 12 control and 10 ACh samples from 4 experiments; unpaired t-test: *p<0.05). Values were corrected for nonspecific uptake in the presence of FCCP (inhibitor of vesicular transporters). ACh: Acetylcholine; cKO: Conditional knockout; IPN: Interpeduncular nucleus; SV: Synaptic vesicle;VTA: Ventral tegmental area

Therefore we assayed locomotor activity in response to single injections of nicotine (0.32, 0.65 and 1.5 mg/kg). Baseline activity of ChAT-cKO mice was comparable to wt (*Figure 11A, B*); however nicotine-induced hypolocomotion was significantly less pronounced in ChAT-cKO than in wt mice (*Figure 11C*), indicating that ChAT-cKO mice show reduced sensitivity to nicotine.

Tolerance to nicotine and withdrawal severity are both critical for the development and maintenance of dependence in humans and animal models (*Changeux, 2010*; *West et al., 1989*). Nicotine

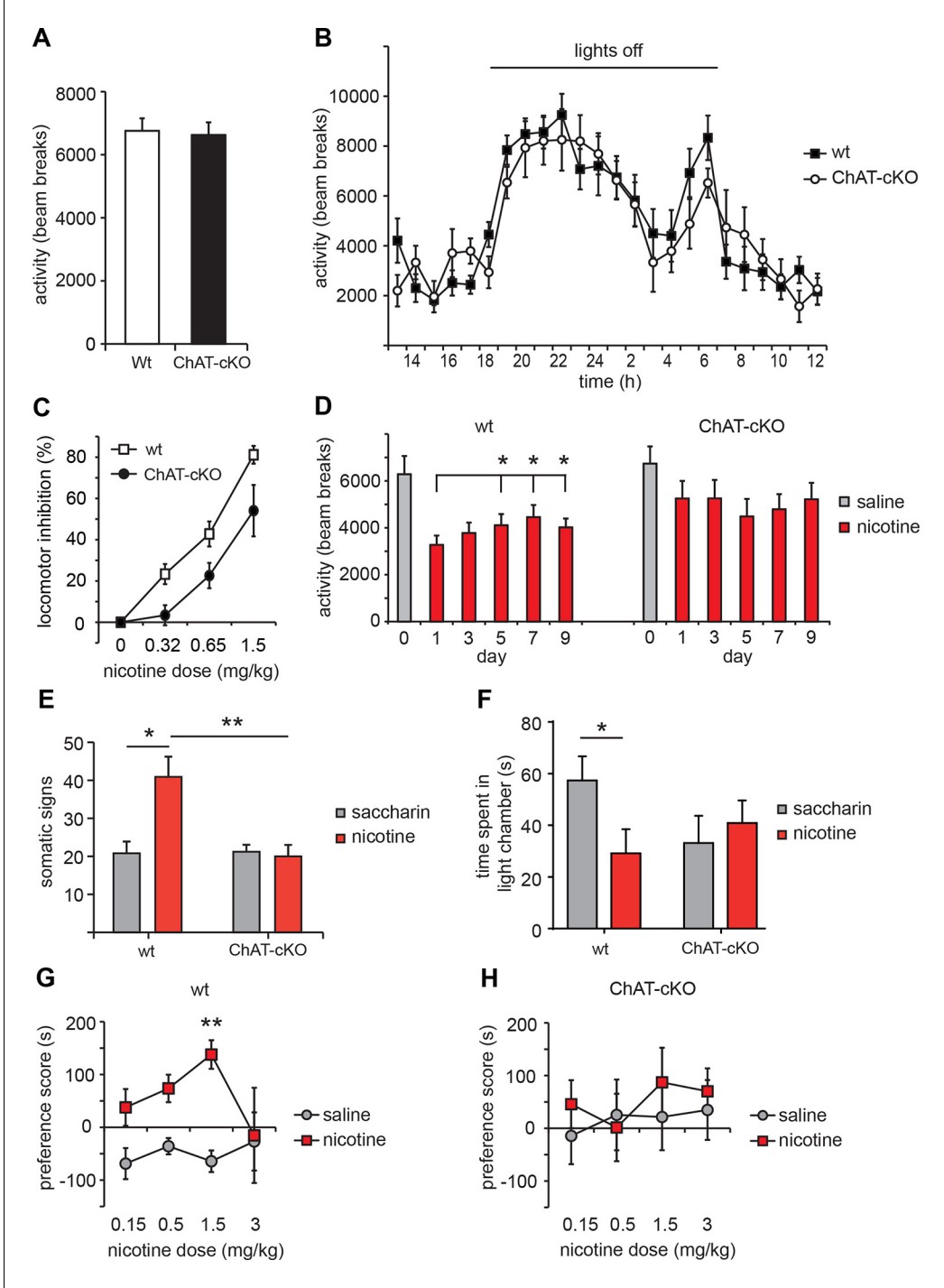

**Figure 11.** ChAT-cKO mice are insensitive to nicotine-conditioned reward and show no signs of nicotine withdrawal. (A) Basal activity, measured as locomotion in a novel environment is similar in ChAT-cKO and wt mice (n = 9 per genotype). (B) Day/night locomotor activity measured as number of beam breaks in 1 hr intervals for 72 hr (averaged to 24 hr) was unchanged in ChAT-cKO (n = 6) compared to wt mice (n = 10). (C) Dose-response of acute effects of nicotine injections on hypolocomotor activity. Nicotine i.p. challenges produced significantly more hypoactivity in wt mice compared to ChAT-cKO mice across all nicotine doses tested (n = 8–10 per genotype and dose; two–way ANOVA for genotype: p<0.001). (D) Daily i.p. injection of 0.65 mg/kg nicotine induced tolerance behavior in wt but not in ChAT-cKO mice: hypolocomotion was significantly attenuated in wt after 5 days of treatment compared to the first day of administration (n=6 per genotype, one–way-ANOVA days 1–9: wt $F_{(2.446)}$ = 4.963, p<0.05 and ChAT-cKO $F_{(1,548)}$ = 0.988, n.s.; paired t-test *p<0.05) but ChAT-cKO maintained similar locomotor activity at consecutive days of nicotine treatment. (E-F) Nicotine withdrawal was measured by quantification of (E) somatic signs (including scratching, rearing, head nods, body shakes and grooming) and (F) anxiety-like behavior (as time spent in dark versus light chamber). Signs of nicotine withdrawal were measured in mice drinking saccharin-sweetened water

*Figure 11 continued on next page*

*Figure 11 continued*

(somatic tests: n=5 per genotype, affective tests: n=8 for wt and n=14 for cKO) or saccharin-sweetened water containing nicotine (somatic tests: n=11 for wt, n=9 for ChAT-cKO; affective: n=11 for wt, n=13 for cKO) for 6 weeks. Precipitation of withdrawal by injection of the nicotinic antagonist mecamylamine elicited somatic and affective signs of withdrawal in wt mice treated with nicotine compared to the saccharin-treated mice (unpaired t-test: *p < 0.05). ChAT-cKO mice treated with saccharin spent less time in the light camber than wt mice treated with saccharin, but this difference was not statistically significant (unpaired t-test; p=0,14). ChAT-cKO mice showed no significant signs of either somatic or affective nicotine withdrawal since there were no differences between the nicotine and control groups (unpaired t-test: **p < 0.01, *p<0.05). (G,H) Nicotine place preference was measured by pairing nicotine or saline injections to different chambers for 3 consecutive days and measuring the time spent in either chamber on the test day. Wt (G) but not ChAT-cKO (H) mice showed robust conditioned place preference after pairing the environment with 1.5 mg/kg nicotine. Each point represents the mean ± S.E.M. of 6–10 mice per group (paired t-test: nicotine vs. saline compartment, **p<0.01; two–way matched ANOVA for substance: **p=0.007 in wt). cKO: Conditional knockout.

tolerance can be observed in mice as a decreased response to repetitive exposure to the same amount of nicotine (*Tapper et al., 2007*; *Tapper et al., 2004*). To test whether this behavior was affected, acute i.p. injections of 0.65 mg/kg nicotine, a concentration that produced approximately 50% activity reduction in wt (*Figure 11C*), were given once daily and activity changes were monitored. Wt mice but not ChAT-cKO became progressively less responsive to daily injections of the drug and locomotor depression was significantly attenuated after 5 days compared to the first day of administration (*Figure 11D*). These data show that ChAT-cKO mice do not develop tolerance induced by repetitive administration of nicotine. We next measured withdrawal responses after cessation of chronic nicotine administration via drinking water over 6 weeks, starting at a concentration of 32.5 μg/ml in the first week, increasing to 65 μg/ml in the second week and to 162.5 μg/ml in the remaining 4 weeks (*Gorlich et al., 2013*; *Salas et al., 2009*). Withdrawal largely prevents success in quitting smoking in humans (*Changeux, 2010*; *West et al., 1989*) and manifests as a collection of physical and affective symptoms that are also observed in mice (reviewed in (*Antolin-Fontes et al., 2015*)). Physical signs of withdrawal ('somatic signs') include scratching, rearing, jumping, head nods, and body shakes (*Damaj et al., 2003*; *Grabus et al., 2005*), while affective withdrawal is measured as anxiety-like behaviors (*Damaj et al., 2003*; *Kenny and Markou, 2001*). Wt mice treated with nicotine exhibited significantly more somatic (*Figure 11E*) and affective (*Figure 11F*) signs of withdrawal compared to wt mice drinking saccharin alone. In contrast, ChAT-cKO mice did not display altered signs of withdrawal in the treated versus untreated group (*Figure 11E, F*). Interestingly, ChAT-cKO mice spent less time than wt in the light chamber at baseline, although this difference was not significant (*Figure 11F*). These results demonstrate that local elimination of ACh release by MHb neurons to IPN neurons prevents the development of nicotine withdrawal behaviors.

To further study the motivational properties of nicotine, we performed the conditioned place preference (CPP) test at different nicotine concentrations. Robust preference for the nicotine-paired environment was seen in wt mice after three conditioning sessions with 1.5 mg/kg nicotine compared to the saline-paired environment (*Figure 11G*). In agreement with previous reports (*Jackson et al., 2010*), a reinforcing response was not induced at very low and very high nicotine doses (0.15 and 3 mg/kg), resulting in a bell-shaped, dose-dependent CPP curve for wt mice. In contrast, ChAT-cKO mice did not display CPP at any nicotine concentration tested (*Figure 11H*), demonstrating insensitivity to nicotine-mediated place preference in ChAT-cKO mice. Taken together, these results reveal that nicotine-dependent behaviors such as nicotine-conditioned reward and withdrawal are controlled by the release of ACh from MHb neurons.

## Discussion

Our brain is controlled by precisely regulated quantities of neurotransmitters and hormones that shape plasticity and control responses to external and internal stimuli. Alterations in the balance of neurotransmitters influence our capacity to cope with addiction, depression and, in the most pronounced cases, can contribute to psychiatric disorders. The studies presented here reveal that ACh production at MHb-IPN synapses is essential for the establishment of nicotine dependence, and that local ACh action plays an important role in withdrawal responses. We demonstrate that loss of ACh selectively in habenular neurons results in a reduction in the content of glutamate in synaptic vesicles isolated from the IPN, supporting the recent finding that MHb neurons corelease ACh and

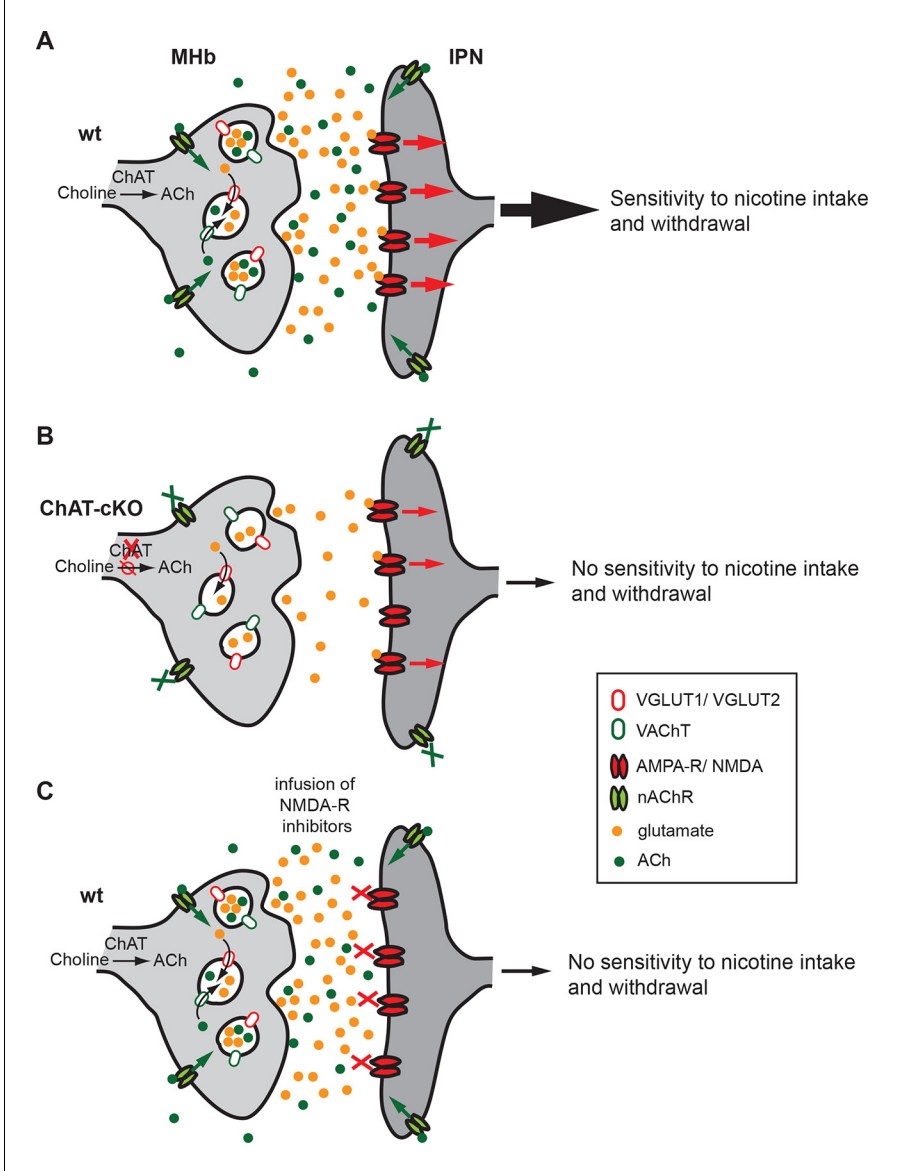

**Figure 12.** Cholinergic and glutamatergic cotransmission from habenular neurons to the IPN is essential for development of nicotine dependence. (A–C) Schemes illustrating the co-transmission of glutamate and ACh in three situations: in wt mice (**A**), in ChAT-cKO mice (**B**), and in mice injected with glutamate receptor blockers (**C**). (**A**) The vesicular transporters VACHT and VGLUT1/2 are colocalized in the great majority of synaptic vesicles at axonal terminals of cholinergic habenular neurons in the IPC. ACh synthesized by cholinergic habenular neurons promotes the uptake of glutamate into SVs by vesicular synergy between neurotransmitter transporters. The simultaneous corelease of ACh and glutamate activates presynaptic nAChR and postsynaptic nAChRs and glutamate receptors and is essential for tapering nicotine intake and producing withdrawal symptoms upon nicotine cessation. (**B**) In the absence of ACh in ChAT-cKO both presynaptic facilitation of glutamate release and vesicle glutamate quantal content are reduced leading to reduced glutamate release at ChAT-cKO synapses and consequently insensitivity to nicotine consumption and withdrawal effects. (**C**) Similarly, injections of NMDA-R blockers in the IPN reduce glutamatergic signaling and diminishes sensitivity to nicotine intake and withdrawal (*Fowler et al., 2011* and *Zhao-Shea et al., 2013*). ACh: Acetylcholine; cKO: Conditional knockout; IPC: Central IPN; IPN: Interpeduncular nucleus; nAChR: Nicotinic acetylcholine receptor; SV: Synaptic vesicle.

glutamate (*Ren et al., 2011*). We provide functional evidence for a dual role of ACh in the IPN: first, the facilitation of vesicular glutamate uptake into SVs; and second, the modulation of transmitter release through activation of presynaptic nAChRs. Both mechanisms influence the excitability of IPN neurons (*Figure 12*), thus accounting for the lack of nicotine dependence in ChAT-cKO mice.

## Vesicular synergy and quantal size

The cotransmission of two neurotransmitters is not uncommon in the nervous system (for reviews see [*Hnasko and Edwards, 2012*; *El Mestikawy et al., 2011*]). There is physiological evidence for glutamate corelease with dopamine, GABA, adrenaline, noradrenaline, serotonin, and ACh in specific neuronal populations in the CNS (*El Mestikawy et al., 2011*). This allows a very local control of excitatory and inhibitory balance at individual synapses (*Shabel et al., 2014*). While some studies have documented the presence of distinct transporters in the same SV and determined their functions (*Gras et al., 2008*; *Hnasko et al., 2010*; *Zander et al., 2010*), it is also possible that not all corelease can be recognized by analysis of neurotransmitter enzyme synthesis and packaging transporters (*Tritsch et al., 2014*). It has been shown that the different ionic properties of vesicular transporters can foster physiologically complementing functions depending on the electrical ($\Delta\Psi$, for VGLUT) or pH gradients ($\Delta pH$, for VAChT, VMAT2, VGAT). We show here for the first time that in the absence of ACh, VACHT can no longer promote glutamate co-entry through VGLUT1/2, thus demonstrating synergy between VACHT and VGLUT1/2 in these synapses. Given previous studies of striatal synapses that demonstrated that loss of VGLUT3 results in a decrease of ACh uptake by synaptic vesicles (*Gras et al., 2008*), and our demonstration that loss of ACh affects glutamate uptake by VGLUT1/2, it seems most probable that transport of glutamate and ACh into IPN vesicles occurs by a reciprocal mechanism. Besides MHb neurons and striatal interneurons, other cholinergic neurons also appear to contain a vesicular glutamate transporter or corelease glutamate, including BF neurons (*Allen et al., 2006*; *Gritti et al., 2006*; *Henny and Jones, 2008*), and spinal motor neurons (*Nishimaru et al., 2005*). This is also the case in the marine ray, *Torpedo californica* (*Li and Harlow, 2014*). Cotransmission of ACh and glutamate might therefore be considered the rule rather than the exception raising a fundamental question about the role of cotransmitter release and neurotransmitter synergy at cholinergic synapses and suggesting that vesicular synergy of these transmitters must have added an adaptive advantage.

## Cholinergic habenular synapses: ultrastructure and molecular components

Our studies agree with previous work supporting the coexistence of two modes of transmission: wired and volumetric for glutamate and ACh respectively. In particular, it has been shown that brief stimulation of MHb efferents elicits glutamatergic but not cholinergic responses, and that tetanic stimulation is required to generate slow inward currents mediated by nAChRs (*Ren et al., 2011*). These differences in transmission could be attributed to the existence of a cholinergic synaptic vesicle population distant from the active zone that is released only during high frequency signaling. However, we can exclude this possibility given our demonstration that glutamate and ACh vesicular transporters colocalize in cholinergic SVs (*Figure 6*) and that both cholinergic and glutamatergic SV are found in close proximity and far (>350 nm) from the active zone (*Figure 5*). Therefore the two modes of transmission most likely reflect the different distribution of postsynaptic glutamate and ACh receptors. Indeed, a great deal of evidence suggests that ACh diffuses to pre- and extrasynaptic sites where it modulates the release of other transmitters (*Covernton and Lester, 2002*; *Descarries et al., 1997*; *Descarries and Mechawar, 2000*; *Girod et al., 2000*; *Mansvelder et al., 2009*). The anatomical structure of the IPN, with numerous varicosities along habenular axonal projections that converge throughout the nucleus in a topographically organized manner forming 'en passant' and crest synapses (*Parajuli et al., 2012*), supports this hypothesis. Electron microscopy studies have revealed that these boutons are highly immunoreactive for CHAT (*Lenn et al., 1985*), consistent with our studies indicating that the vesicles visualized in these synapses are cholinergic. Given the high concentration and catalytic power of AChE in the IPN (*Flumerfelt and Contestabile, 1982*; *Franklin and Paxinos, 2008*; *Quinn, 1987*), the pre-terminal localization of synaptic boutons in 'en passant' synapses along habenular axons (*Parajuli et al., 2012*), and our results showing colocalization of ACh and glutamate transporters in the same SV populations, it seems plausible that high frequency stimulation of MHb terminals is necessary to release sufficient ACh to reach nAChRs in the IPN (volume transmission) while brief stimulation can elicit fast postsynaptic glutamatergic responses (wired transmission).

## Prominent role of the MHb-IPN circuit in nicotine dependence

Several nAChR subtypes in the MHb-IPN pathway are key in the control of nicotine consumption. Receptors containing the β4 and/or α5 subunit contribute to the activation of this tract during nicotine intake and are critical for the aversive (reward-inhibiting) effects of high-dose nicotine (*Fowler et al., 2011*; *Frahm et al., 2011*), while α2, α5 and β4-subunits play a major role in nicotine withdrawal (*Salas et al., 2004*; *Salas et al., 2009*). Less is known about the contribution of the MHb-IPN pathway to the reinforcing effects of nicotine. Intracerebroventricular injection of the α3β4 nAChR blocker conotoxin AuIB, resulted in attenuated reward and withdrawal from nicotine (*Jackson et al., 2013*). However, although α3β4 nAChR subunits coassemble with α5 nAChRs in about 15–35% of MHb-IPN neurons (*Grady et al., 2009*), the reward-enhancing properties of nicotine are not altered in α5-KO mice (*Fowler et al., 2011*; *Fowler et al., 2013*; *Jackson et al., 2013*). Given the large variety of nAChR subtypes in this circuit and their differential distribution on glutamatergic and GABAergic pre- and postsynaptic terminals, the advantage of local removal of the endogenous ligand ACh to understand the impact of nicotine on this tract is apparent. Thus, using ChAT-cKO mice we were able to demonstrate for the first time that ACh in the MHb-IPN circuit is crucial for the establishment of dependence-related behaviors, including reward and tolerance to nicotine.

Presynaptic nAChRs at habenular synapses facilitate glutamate and ACh release (*Girod et al., 2000*; *McGehee et al., 1995*), and postsynaptic nAChRs contribute to an increased excitatory response upon activation. However, it appears that GAD-immunoreactive cells constitute the largest population of neurons in the IPN (*Kawaja et al., 1989*) and activation of α4β2 nAChRs located on GABAergic interneurons can trigger spike discharge and GABA-release (*Covernton and Lester, 2002*; *Lena et al., 1993*), suggesting that ACh and nicotine modulate both glutamatergic and GABAergic synaptic transmission in the IPN. Importantly, repeated exposure to nicotine leads to inactivation of nAChRs, thus altering the modulatory effects of the endogenous transmitter ACh. Given that nicotine increases GABA release in the IPN and that continuous nicotine exposure desensitizes α4β2 nAChRs considerably more than α7 and α3β4 nAChRs (for review see [*Giniatullin et al., 2005*]), it is possible that persistent desensitization of α4β2 nAChRs on GABAergic neurons in the IPN increases the net excitatory output of this tract, as previously demonstrated in the ventral tegmental area (VTA) (*Mansvelder et al., 2002*). The lack of dynamic changes in nAChR activity on inhibitory neurons in the absence of ACh, and subsequent altered stimulation of efferent targets, could thereby contribute to the insensitivity of ChAT-cKO mice to nicotine-mediated reward and tolerance. Descending projections from the IPN include the dorsal raphe (DR) nucleus and LDTg. Both nuclei exchange reciprocal information and provide glutamatergic (*Geisler and Wise, 2008*) and cholinergic (*Maskos et al., 2005*) input to the VTA. Therefore, loss of cholinergic communication in the MHb-IPN axis, indirectly influencing activation of the mesocorticolimbic dopaminergic system, fails to induce nicotine-dependent behaviors, including reward, tolerance and withdrawal in ChAT-cKO animals.

## Genetic predisposition to addiction

The reinforcing properties of nicotine together with the negative withdrawal symptoms that develop upon nicotine cessation are the main reasons for the maintenance of the smoking habit in spite of its known deleterious consequences (*De Biasi and Dani, 2011*). Genome-wide association studies have identified single nucleotide polymorphisms (SNPs) as risk factors for nicotine dependence and lung cancer in genes encoding nAChR subunits (*CHRNB3-CHRNA6* and *CHRNB4-CHRNA3-CHRNA5* clusters), and in nicotine-metabolizing enzymes (*CYP2A6* and *CYP2B6*) (*Bierut et al., 2007*; *Kumasaka et al., 2012*; *Saccone et al., 2009*; *Thorgeirsson et al., 2010*). In addition, variations in the *CHAT* gene have been associated with prospective smoking cessation (*Heitjan et al., 2008*; *Ray et al., 2010*; *Turner et al., 2013*; *Wei et al., 2010*). The studies presented here show that nicotine does not trigger nicotine-dependent behaviors unless the endogenous neurotransmitter ACh is released from MHb neurons to further promote glutamate corelease. In the corticostriatal circuitry, it has been proposed that glutamate homeostasis underlies drug-seeking behavior (*Kalivas, 2009*). After repetitive drug use, deregulation of this homeostasis increases the release of glutamate during drug relapse (*Kalivas, 2009*). Similarly, in the MHb, re-exposure to nicotine after abstinence increases the pacemaking of MHb neurons (*Gorlich et al., 2013*) and subsequent corelease of ACh

and glutamate from habenular axonal terminals. Although it is known that the main action of nAChRs is presynaptic in the CNS and postsynaptic at the neuromuscular junction, no studies have addressed the role of ACh in this mechanism. The studies presented here not only agree with the conclusion that nicotine-induced behaviors act via presynaptic receptors that regulate glutamate release but also show that nicotine is not sufficient to induce dependence unless ACh is released and promotes further release of glutamate. Given that other cholinergic neurons beside MHb neurons corelease glutamate, and our findings demonstrating that ACh controls the quantal size and release frequency of glutamate, it is possible that the synergistic functions of ACh and glutamate may be generally important for modulation of cholinergic circuit function and behavior and that genetic variations in *CHAT* may influence several aspects of glutamatergic transmission in the addiction process.

## Material and methods

### Animals

*Chat$^{flox/flox}$* mice (*Misgeld et al., 2002*) were crossed to *Kiaa1107-Cre* mice (founder KJ227 GEN-SAT) (*Gong et al., 2003*). *Kiaa1107* is annotated as *A830010M20Rik* by GENSAT. Cre-positive mice homozygous for the floxed allele were used as ChAT-cKO. Cre-negative littermates were used as wt control animals. *Gt(ROSA)26Sor$^{tm1(EYFP)Cos}$* (The Jackson Laboratory, Sacramento, CA) reporter line was crossed to *Kiaa1107-Cre* line. Adult rats (Charles River Laboratories, Wilmington, MA) were used for synaptic vesicle isolation experiments. Mice were housed with *ad libitum* access to food and water in room air conditioned at 22–23°C with a standard 12 hr light/dark cycle, maximal five animals per cage. All procedures were in accordance with ethical guidelines laid down by the local governing body.

### Drugs

(-) Nicotine hydrogen tartrate salt and mecamylamine hydrochloride were purchased from Sigma-Aldrich (St. Louis, MO). Drugs were dissolved in 0.9% saline and administered intraperitoneally (i.p.) in a volume of 100 µl per 10 g body weight. Nicotine concentrations refer to the free base.

### Brain immunohistochemistry and quantification analysis

Immunohistochemistry was performed as described by (*Frahm et al., 2011*) in adult mice (3–4 weeks) except in *Figure 3* (p6-p7 mice). The primary antibodies used were goat polyclonal anti-CHAT (1:1000, EMD Millipore, Germany), mouse monoclonal anti-Tyrosine hydroxylase (1:2000, Sigma-Aldrich), guinea pig polyclonal anti-VGLUT1 (1:1000, Synaptic Systems, Germany), guinea pig polyclonal anti-VGLUT2 (1:1000, Synaptic Systems), rabbit polyclonal anti-VACHT (1:500, Synaptic Systems) and rabbit polyclonal anti-EGFP (1:1000, Invitrogen, Waltham, MA). The sections were incubated with primary antibodies overnight at 4°C. After incubation with secondary antibodies sections were washed, mounted on slides and coverslipped in immu-mount (Thermo Scientific, Waltham, MA). Heat-mediated antigen retrieval, 15 min at 95°C in citric acid (pH 6.0), was performed prior to incubation with the choline acetyltransferase (CHAT) antibody to enhance immunostaining. Fluorescent signals were detected using a confocal laser scanning microscope (Leica SP5 or Zeiss LSM700, Germany) and Biorevo fluorescent microscope (Keyence, Japan).

Image J was used for quantification analysis. The number of CHAT and EYFP cells per section was quantified in *Kiaa1107-Cre* crossed to *Gt(ROSA)26Sor$^{tm1(EYFP)Cos}$* mice. 8 to 16 sections from three different mice were used for analysis. The number of CHAT positive neurons per section was quantified in wt and ChAT-cKO mice. 13 to 22 sections from three different mice per genotype were used for analysis.

### Colocalization analysis

Images used for colocalization analysis were acquired with a 40× oil immersion objective on a Zeiss LSM700 confocal microscope. The analysis was done as described in (*Broms et al., 2015*). Briefly, the detection pinhole was set to one Airy unit and the channels were captured in sequence. The intensity gain was adjusted for each channel before capture and the intensity range of the images was left untouched to preserve linearity. Colocalization analysis was performed with the Coloc 2 plugin of the Fiji image processing package. Background was eliminated by median subtraction

(*Dunn et al., 2011*). Manders' colocalization coefficients (M1 and M2), which are proportional to the number of colocalizing pixels in each channel relative to the total number of pixels, were calculated (*Manders et al., 1993*). M1 or M2 > 0.55 indicates colocalization (*Zinchuk and Grossenbacher-Zin-chuk, 2014*). In this study, only M1 is presented and refers to the proportion of pixels with VGLUT1 immunoreactivity that colocalize with either VACHT or CHAT. Costes' test for statistical significance was used to determine that the colocalization coefficients obtained were not due to random effects (*Costes et al., 2004*). This test creates random images by shuffling blocks of pixels of one channel, measuring the correlation of this channel with the other (unscrambled) channel of the same image. The test was performed 100 times per image and the resulted P value indicates the proportion of random images that have better correlation than the real image. A P-value of 1.00 means that none of the randomized images had better correlation. The M1 and P value below panels 1–4 of *Figure 4* were calculated from one individual image per IPN subnucleus.

## Slice preparation and electrophysiological recordings

For recordings of evoked excitatory postsynaptic currents (eEPSCs) from MHb and IPN somata, slices were cut at 350 µm from adult mice, on a Microm HM 650 V (Thermo Scientific). Slices were kept submerged at room temperature in oxygenated (95% $O_2$ and 5% $CO_2$) artificial cerebrospinal fluid containing (mM): 125 NaCl, 2.5 KCl, 1.3 $MgCl_2$, 2 $CaCl_2$, 1.25 $KH_2PO_4$, 11 glucose and 26 $NaHCO_3$ (pH 7.4; osmolarity 310 mosmol l−1). The internal pipette solution contained (mM): 120 potassium gluconate, 2 $MgCl_2$, 0.5 $CaCl_2$, 5 EGTA and 10 Hepes (pH 7.3; resistance 4–5 MΩ). Nicotine was locally applied by pressure (8 – 10 psi, 100 ms) with pipettes similar to the recording pipette using a pressure regulator (PR-10; ALA Scientific Instruments, Farmingdale, NY) controlled with a trigger interface (TIB 14S; HEKA Electronics, Germany). The pipette was moved within 15–20 µm of the recorded cell using a motorized micromanipulator (LN mini 25, control system SM-5; Luigs & Neumann, Germany) for drug application and retracted at the end of the puff to minimize desensitization.

For electrophysiological recordings of mEPSCs in the IPN, adult mice were sacrificed by cervical dislocation and brains were dissected in chilled (4°C) artificial cerebrospinal fluid (ASCF) containing (in mM): 87 NaCl, 2 KCl, 0.5 $CaCl_2$, 7 $MgCl_2$, 26 $NaHCO_3$, 1.25 $NaH_2PO_4$, 25 glucose, 75 sucrose, bubbled with a mixture of 95% $O_2$/5% $CO_2$. 250 µm coronal IPN containing slices were cut with a VT1200S vibratome (Leica), preincubated for 30 min at 37°C and then transferred to the recording solution containing (in mM): 125 NaCl, 2.5 KCl, 2 $CaCl_2$, 1.3 $MgCl_2$, 26 $NaHCO_3$, 1.25 $NaH_2PO_4$, 10 glucose, 2 sodium pyruvate, 3 myo-inositol, 0.44 ascorbic acid, bubbled with a mixture of 95% $O_2$/5% $CO_2$. Slices rested in this recording solution at room temperature for at least one hour before they were transferred to the recording chamber and perfused with ACSF containing 1 µM TTX, 100 µM Picrotoxin, and 1 µM CGP 55,845 hydrochloride (all from Tocris, UK). Electrophysiological responses were recorded with an EPC 10 patch-clamp amplifier and PatchMaster and FitMaster software (HEKA Elektronik). Cells were patch-clamped at -70 mV for 10 min, before 100 µM ACh or 250 nM nicotine tartrate (Sigma Aldrich) was washed in for 5 min. Spontaneous mEPSCs were analyzed for the one minute before ACh or nicotine application (baseline), and for the last minute of drug application with Mini Analysis (Synaptosoft, Decatur, GA). Patch pipettes had resistances of 4–8 MΩ when filled with a cesium-based solution containing (in mM): 105 CsMeSO, 30 CsCl, 10 Hepes, 10 phosphocreatine, 2 ATP-$Mg^{2+}$, 0.3 GTP (pH adjusted to 7.2 with CsOH). In recordings of the IPL, 3–5 mg/ml Biocytin (Sigma Aldrich) was included in the intracellular recording solution. After recordings, slices were fixed with 4% PFA for 30–45 min, and incubated with a streptavidin conjugated Alexa Fluor 488 dye (Life technologies, Guilford, CT) to label biocytin filled neurons, and a rat primary antibody for substance P (Santa Cruz, Dallas, TX) to label the IPL. After incubation with secondary antibodies, fluorescent signals were detected using a confocal laser-scanning microscope (Zeiss). Results are presented as means ± SEM.

## Western blotting

The MHb and IPN were dissected from adult ChAT-cKO and wt mice (n = 3 per genotype), collected in 1 ml of lysis buffer (50 mM Na phosphate pH 7,4, 1 M NaCl, 2 mM EDTA, 2 mM EGTA and protease inhibitor cocktail) and immediately homogenized. The homogenates were processed as described in (*Frahm et al., 2011*). Primary antibodies used were goat polyclonal anti-CHAT

(Millipore), mouse monoclonal anti-α-Tubulin (Sigma-Aldrich), rabbit polyclonal anti-nAChR β4 (gift from Dr. Cecilia Gotti) and rabbit polyclonal anti-GluR1 (Millipore).

## Electron microscopy

### Fixation

Adult mice were perfused transcardially with 0.9% NaCl for 1 min followed by fixative containing 4% PFA in 0.1M sodium cacodylate buffer (pH 7.4) for 10 min. The brains were kept in the fixative overnight at 4°C and sectioned coronally with a vibrating microtome (VT1000S vibratome, Leica) at 50 μm thickness the following day.

### Pre-embedding nanogold immunolabeling

IPN coronal sections were incubated in blocking solution (3% BSA, 0.1% saponin) in the buffer for 2 hr at room temperature. The sections were then single labeled at 4°C overnight with the following primary antibodies diluted in blocking solution: rabbit polyclonal serum against the vesicular acetylcholine transporter (VACHT; 1:10,000) (*Gilmor et al., 1996*) and guinea pig polyclonal antibody against the vesicular glutamate transporter 1 (VGLUT1; 1:3,000) (135 304; Synaptic Systems). The following day, the sections were incubated in secondary antibody diluted 1:100 for 1 hr at room temperature (anti-Rabbit: 2003, and anti-Guinea pig: 2054, Nanoprobes, Yaphank, NY ). The sections subsequently underwent silver enhancement (HQ Silver Enhancement 2012, Nanoprobes), Gold Toning using a 0.1% solution of gold chloride (HT1004, Sigma Aldrich) and fixed overnight in 2.5% glutaraldehyde at 4°C. The control experiment was done by following the same procedure except for omitting the primary antibody and incubating with blocking solution instead. The IPN was excised and postfixed with 1% osmium tetroxide in the buffer in the presence of 1.2% potassium ferrocyanide for 1 hr on ice. Sections underwent *en bloc* staining with 1% uranyl acetate for 30 min, dehydration in a graded series of ethanol, infiltration with Eponate 12™ Embedding Kit (Ted Pella, Redding, CA), embedding with the resin and polymerization for 48 hr at 60°C. Series of 90 nm ultrathin sections were cut and analyzed on a JEOL JEM-100CX at 80 kV using with the digital imaging system (XR41-C, Advantage Microscopy Technology Corp, Denver, MA).

### High pressure freezing and freeze substitution

Isolated IPN from 50 μm vibratome sections were applied to a high pressure freezer (EM PACT2, Leica) in cryoprotectant (30% BSA) and transferred to a freeze substitution unit (EM AFS2, Leica). They were incubated with a substitution medium, containing 1% uranyl acetate, 95% acetone and 5% water, at -85°C, washed in acetone, infiltrated with Lowicryl HM20 and embedded in the resin at -45°C. Ultrathin sections (90 nm) were cut and processed for post-embedding immunolabeling.

### Post-embedding immunolabeling

Post-embedding immunolabeling was carried out on 90 nm ultrathin sections mounted on carbon film 200 mesh nickel grids (Electron Microscopy Sciences, Hatfield, PA). Sections were incubated in blocking solution containing 3% BSA, 0.1% saponin in 20 mM Tris (pH 7.5) for 2 hr at room temperature, incubated with single or double primary antibodies (rabbit polyclonal serum anti-VACHT; 1:800) (*Gilmor et al., 1996*) and/or guinea pig polyclonal anti-VGLUT1; 1:18,000 (135 304; Synaptic Systems)) diluted in blocking solution overnight at 4°C, rinsed in 20 mM Tris (pH 7.5) for 30 min, incubated with appropriate secondary antibodies (e.g., anti-rabbit IgG conjugated to 6 nm gold particles and anti-guinea pig IgG conjugated to 12 nm gold particles (Jackson ImmunoResearch, West Grove, PA) diluted 1/20 in 20 mM Tris (pH 7.5) containing 0.5% BSA and 0.1% saponin for 1 hr at room temperature. After washing with the buffer and water, the sections were stained with 1% osmium tetroxide and lead citrate and examined in a JEOL JEM-100CX microscope.

### Quantitative analysis

Quantification was performed on preembedding nanogold labeled synaptic terminals (at 20000 magnification) only when active zone, synaptic cleft and vesicles were clearly presentable (VACHT, n=19 terminals; VGLUT1, n=21 terminals from two wt mice). In a separate analysis, the distance between a VGLUT1/VACHT-gold particle and the nearest VGLUT1/VACHT-gold particle was

measured using electron micrographs with postembedding double immuno gold labeling (n=109 terminals from 2 mice). Both measurements were done with ImageJ.

## Immunoisolation of synaptic vesicles

Ten mice per genotype (ChAT-cKO and wt littermates) were used to dissect IPN brain samples. Brain tissue was resuspended in ice-cold buffer (in mM: 320 sucrose, 4 HEPES/KOH/pH 7.4, 1 PMSF, protease inhibitor for mammalian tissue (Pi) 1:1000 (Sigma-Aldrich) and homogenized (10 strokes at 900 rpm (homogenizer clearance: 0.1–0.15 mm, Wheaton, UK)). Nuclei were discarded by centrifugation and the supernatant mainly containing SVs was diluted 1:10 in lysis buffer (ddH$_2$O, in mM:10 HEPES/KOH/pH 7.4, 1 PMSF, Pi 1:1000) for osmotic shock and processed with three strokes at 2000 rpm using a Wheaton homogenizer to obtain SVs in suspension. Immunoisolation of SV was performed using paramagnetic beads (Dynabeads: for monoclonal antibodies: Pan Mouse IgG, for polyclonal rabbit antibodies: M-280 sheep anti-rabbit, Life Technologies) were first coated with the respective primary antibodies suspended in coating buffer [PBS, pH 7.4, 0.1% BSA (w/v)], supplemented with 0.5 – 1 µg of IgG per 10$^7$ Pan Mouse IgG beads or 2 x 10$^7$ M-280 sheep anti-rabbit beads and rotated for 2 hr at 4°C followed by four washing steps. Immunoisolation was performed overnight at 4°C under rotation using a suspension of the coated beads and S1shock fractions diluted with incubation buffer [PBS, pH 7.4, 2 mM EDTA, and 5% BSA (w/v)] to yield a ratio of 50–75 µg protein to 1.4 x10$^7$ beads (Pan Mouse IgG) or 3.9 x 10$^7$ beads (M-280). SV bound to beads were washed three times in incubation buffer, and three times in coating buffer. Bead-bound SV were finally dissolved in sample buffer (*Takamori et al., 2000*; *Zander et al., 2010*). Beads without primary antibodies (for monoclonal immunoisolation) or with normal rabbit IgG (Santa Cruz, for isolation using polyclonal rabbit antibodies) were subjected to the same procedures and served as control for non-specifically bound material. The protein pattern of isolated SV was analyzed by SDS-PAGE and Western blot. Antibodies used for Western blot were rabbit polyclonal anti VGLUT1 or VAChT (Synaptic Systems).

## Glutamate uptake

Glutamate uptake was performed using a SV-enriched fraction (S1 hyposmotically shocked and spun down as pLP2) from 10 rat IPN. Briefly, membranes of S1shock fraction (see above) were removed (20 min at 29,000 x g) and SV were pelleted (30 min at 350,000 x g (TLA-100.4 rotor, Beckman Coulter, Jersey City, NJ)) resulting in pLP2 fraction. Uptake was performed in K-gluconate buffer (in mM: 146 K-gluconate, 4 KCl, 20 1,4-piperazinediethanesulfonic acid, 4 EGTA, 2.9 MgSO$_4$ (corresponding to 1 mM free Mg$^{2+}$), 2 Na-ATP, adjusted to pH 7.0 with KOH) by adding 1 µM neostigmine, 49.5 µM K-glutamate and 0.5 µM [3,43H]-L-glutamic acid (Hartmann Analytic GmbH, Germany) to the SV suspension. Non-specific uptake was performed by adding 60 µM FCCP also used for background correction. Additives were given 10 min before the uptake was started at 25°C. The reaction was stopped after 10 min by adding ice-cold buffer containing the same ionic constituents used during the uptake. Non-bound radioactivity was removed by centrifugation at 435,000 x g (TLA-120.1 rotor, Beckman Coulter) for 10 min, and the pellets were washed three times. Radioactivity was measured by liquid scintillation counting (LS 6500, Beckman Coulter). (*Zander et al., 2010*; *Preobraschenski et al., 2014*).

## Locomotor behavior and tolerance

The MoTil system (TSE Systems, Germany) was used to monitor open-field behavior of 2–3 month old male mice in complete darkness (20 x 40 x 40 cm black box). Single events represented disruption of two distinct infrared photobeams 3 cm apart in the cage. After injection of saline or different concentrations of nicotine, the number of beam breaks was measured for 15 min. Locomotor inhibition was calculated by normalizing to basal activity values (beam breaks nicotine x 100% / beam breaks saline). For measurement of tolerance behavior, mice were given daily i.p. injections of 0.65 mg/kg nicotine (total of 9 days), starting with saline at day 0 as control activity. Activity was monitored in the open field box every second day for 15 min immediately after injections.

Home cage activity of single housed animals was measured for 3 days and averaged to 24 hr.

## Conditioned place preference

A rectangular three-compartment box separated by removable doors (TSE-systems) was used. The center compartment (6 x 15 x 20 cm) is grey with a polycarbonate smooth floor. The choice compartments (17 x 15 x 20 cm) have different visual and tactile cues. Time spent in each compartment was measured with photobeam detectors. During the preconditioning phase (day1) male mice (8–12 weeks old) were placed into the center compartment with closed doors. After 2 min of habituation, doors opened automatically and mice were allowed to explore the three compartments freely for 15 min. The pre-conditioning session was used to determine baseline responses and less preferred compartments were paired with nicotine. Mice exhibiting a strong preference for one side (>150 s) in the pre-conditioning session were excluded. During the conditioning phase (day 2–4), mice were given a saline injection in the morning and a nicotine injection in the afternoon and were confined to either one side or the other of the conditioning box for 20 min. On the preference test (day 5), the doors between the compartments were opened after mice habituated for 2 min in the central chamber and were then allowed to move freely in the three compartments for 15 min. Preference score was measured as time spent in nicotine-paired or saline-paired compartment at day 5 – day 1.

## Nicotine treatment and withdrawal tests

Single housed male adult mice were treated with nicotine via the drinking water over 6 weeks, starting at a concentration of 32,5 μg/ml in the first week, increasing to 65 μg/ml in the second week and to 162,5 μg/ml in the remaining 4 weeks (*Gorlich et al., 2013*; *Salas et al., 2009*). To minimize taste aversion, 2% saccharin was added to both, treatment and control groups. Withdrawal was precipitated by i.p. injection of 2 mg/kg mecamylamine. To measure somatic signs of withdrawal, mice were placed into their home cage immediately after injection and videotaped for 20 min. During this time, following events were scored as somatic signs of withdrawal: body tremors, head nods, paw tremors, grooming, genital licks and scratching. To measure affective signs of withdrawal we employed a light-dark box by placing a dark enclosure with a doorway on one side of the open field boxes (Accuscan & Omnitech Electronics, Columbus, OH). We employed another set of mice (treated with saccharin and nicotine) different from the one used for somatic withdrawal. Withdrawal was precipitated by i.p. injection of 2 mg/kg mecamylamine. Immediately after injection, mice were placed in the dark chamber and allowed to move freely between both chambers for 5 min. Time spent in each chamber was recorded.

## Statistical analysis

Statistical analysis was performed with GraphPad Prism 6.0. Unpaired two-tailed Student t-tests were used for analyzing most of the data, except when two-way analysis of variance with ANOVA or paired two-tailed Student t-tests are indicated. Results are presented as means ± S.E.M.

# Acknowledgements

We thank Dr. Kunihiro Uryu and Dr. Nadine Soplop (Rockefeller University, New York) for EM expertise and technical assistance. We thank Awni Mousa for statistical analyses, Julio Santos-Torres, Sylvia M. Lipford and Will Heintz for technical help and Dr. Jessica L. Ables for critical discussions (Rockefeller University, New York). We thank Dr. Joshua R Sanes (Harvard University, Cambridge, MA) for the conditional *Chat* mouse, Dr. Allan Levey (Emory University, Atlanta) for the VACHT antibody and Dr. Cecilia Gotti (University of Milan) for the β4 nAChR antibody. This work was supported by the Helmholtz Association (IIT) and the Deutsche Forschungsgemeinschaft (DFG): grant GO 2334/1-1 (A.G) and AH67/7-1 (GAH), HHMI and NIH/NIDA: grant 1P30 DA035756-01 (II-T).

# Additional information

## Funding

| Funder | Grant reference number | Author |
| --- | --- | --- |
| National Institute on Drug Abuse | P30DA035756 | Ines Ibañez-Tallon |

| Deutsche Forschungsgemeinschaft | GO 2334/1-1 | Andreas Görlich |
| Deutsche Forschungsgemeinschaft | AH67/7-1 | Gudrun Ahnert-Hilger |
| Helmholtz-Gemeinschaft | | Ines Ibañez-Tallon |

The funders had no role in study design, data collection and interpretation, or the decision to submit the work for publication.

### Author contributions

SF, BAF, AG, JFZ, Acquisition of data, Analysis and interpretation of data, Drafting or revising the article; GAH, Analysis and interpretation of data, Drafting or revising the article; IIT, Conception and design, Analysis and interpretation of data, Drafting or revising the article

### Ethics

Animal experimentation: All of the animals were handled according to approved institutional animal care and use committee (IACUC) protocols (#14734-H) of the Rockefeller University and the Max Delbruck Center, Germany. All surgery was performed under sodium pentobarbital anesthesia, and every effort was made to minimize suffering.

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
