## [Decision Letter]

Thank you for resubmitting your work entitled "An Essential Role of Acetylcholine-Glutamate Synergy at Habenular Synapses in Nicotine Dependence" for further consideration at *eLife*. Your revised article has been favorably evaluated by Gary Westbrook (Senior editor) and two reviewers, one of whom, Sacha Nelson, is a member of our Board of Reviewing Editors. The manuscript has been improved but there are some remaining issues that need to be addressed before acceptance, as outlined below:

1) The assumption that two gold particles within a distance of 90 nm label the same vesicle is likely to give an overestimate of the number of vesicles containing both VAChT and VGLUT1, because gold particles on adjacent vesicles can easily be less than 90 nm apart given the close packing. The estimate of double-positive vesicles may be an overestimate, making it difficult to accept the authors' claims that the majority of SVs in these axon terminals contain both transporters. The authors should present a more compelling argument for why 90 nm is not only the maximum distance for labeling in the same vesicle but also the minimum distance for labeling in adjacent vesicles, or should make textual changes acknowledging the colocalization may be lower.

2) Given the similarity of the numbers at the bottom of the subsection “Cholinergic and glutamatergic inputs to IPN subnuclei” it would be appropriate to add the word "slightly", so that the sentence reads "slightly" less overlap in the IPI.

3) The authors' claim that ChAT deletion resulted in the "complete absence" of nicotine-elicited behavioral responses is too strong of an assertion. Nicotine was still able to elicit hypolocomotion in the ChAT-cKO mice, albeit to a lesser degree than WT mice (Figure 11). The authors also did not provide any evidence to distinguish the effects of the loss of ACh release from these MHb neurons from the effects of altered Glu release from the same afferents. Therefore they are unable to convincingly argue that the behavioral effects are also due to the reported loss of the vesicle synergy and altered Glu signaling. Experiments that attempted to rescue impaired Glu release resulting from the loss of vesicle synergy (i.e. increasing Glu release from the ventral MHb-IPN projection of the ChAT-cKO mice) could have helped to dissociate the effects resulting from separate outcomes of ChAT deletion from ventral MHb neurons. The authors should make textual changes to moderate the assertion.

[Editors’ note: a previous version of this study was rejected after peer review, but the authors submitted for reconsideration. The previous decision letter after peer review is shown below.]

Thank you for choosing to send your work entitled "An Essential Role of Acetylcholine-Glutamate Synergy at Habenular Synapses in Nicotine Dependence" for consideration at *eLife*. Your full submission has been evaluated by a Senior editor and three peer reviewers, one of whom is a member of our Board of Reviewing Editors, and the decision was reached after discussions between the reviewers. Based on our discussions and the individual reviews below, we regret to inform you the manuscript in its present form will not be considered further for publication in *eLife*.

As you will see all three reviewers were enthusiastic about the work but felt that additional experiments were needed to support the core conclusions of the paper. Since some of these involve breeding additional animals we recognize that it is unlikely that this can be completed in a brief period (i.e. two months). It is the policy of *eLife* to reject manuscripts for which the additional work required would be likely to take several months or longer. However, in this case, we would encourage you to submit a new manuscript that addresses the concerns raised by the authors. We will endeavor to send the manuscript back to the same reviewers, although there is no guarantee that this will be possible.

Although you should address all of the points raised, the reviewers and editors agreed that the following experimental issues were the most essential:

1) The driver strain should be better characterized. At a minimum you should cross the Cre line to a reporter mouse and perform a Chat stain. This would allow you to then quantify both the penetrance and specificity of the line.

2) You need to at least make a better argument for why non-habenular glutamatergic inputs are unlikely and may need to do some additional experiments to shore up this point.

3) You need to better quantify and better support the western blot data, perhaps adding measures of holding current and paired pulse responses.

4) Additional controls are needed for co-localization experiments. This can probably be done simply by reanalyzing the images already acquired.

5) The sample size for Figure 3 should be increased to ~20.

*Reviewer #1:*

In this paper the authors use conditional deletion of CHAT to convincingly demonstrate synergy between ACh and glutamate at MHb synapses as well as showing changes in behavioral correlates of tolerance and withdrawal. These are important and interesting results.

1) My major concern with the paper is that the authors have not demonstrated that this synergy is the only, or even a necessary component of mechanism of the behavioral changes they observe. Given the higher affinity of NMDA receptors than many other glutamate receptors, it is not clear that they would not be activated in the absence of facilitation of glutamate release.

The issue may be somewhat semantic – that is the authors may wish "synergy" in the title and Discussion to apply more broadly to any sort of interaction between ACh and glutamate, as opposed to the vesicular synergy on which they focus for much of the results.

2) A western blot (Figure 3) is not a convincing way to demonstrate either that presynaptic receptors are unchanged or rule out a change in receptor number contributing to altered mini amplitude. If presynaptic receptors were altered this could also contribute to the altered frequency effects of added ACh and nicotine (3L and M).

*Reviewer #2:*

The study described in Frahm et al. investigates the mechanism and functions of glutamate/acetylcholine co-release from MHb inputs onto interpeduncular nucleus (IPN) neurons. The authors have an excellent model for the analysis of MHb neurons as the cre line they describe may be specific for cholinergic neurons in the MHb. They go on to show that deletion of Chat from Cre+ neurons alters mEPSC amplitude in the IPN, and that VAChT and vGlut1 are both expressed on synaptic vesicles in terminals of the IPN. Finally, they show that this mouse line has an attenuated hypolocomotor response to nicotine injection and a lack of nicotine preference in a place preference task. While the authors offer some intriguing results there are major flaws in some of their conclusions. The study would also greatly benefit from a more thorough and quantitative characterization of their MHb/Chat cKO mouse model.

1) While the A830010M20Rik-Cre line may only express Cre in cholinergic neurons of the MHb, it clearly expresses Cre in many other areas of the brain. Quantification of the number of Chat+ neurons in the WT and Chat-cKO in the MHb is required at a minimum.

2) If nicotinic responses are still observable in IPN neurons when nicotine is directly applied (Figure 3), then the authors should also show that cholinergic responses from MHb are abolished in the IPN. An optogenetic experiment similar to that shown in Ren J., et al. 2011 would be appropriate.

3) While the authors show that mEPSC amplitude is decreased in IPN neurons of the Chat-cKO, mEPSC recordings sample from all input that a neuron receives, not just the inputs from MHb. The authors need to show that glutamatergic currents from MHb are affected in the Chat-cKO.

4) In Figure 4 and Figure 5 the authors have no way of identifying if the terminals analyzed are from MHb or somewhere else, yet they state they are from the MHb. The wording should be changed to reflect this, or the authors should find a way to show that these terminals are indeed from the MHb.

5) The authors state that the quantity of vesicular proteins in WT and cKO are similar in Figure 6, however they do not quantify their results or perform any statistical comparison.

6) The physiology experiments in Figure 3 are under sampled and require more cells in order to support the conclusions drawn from the statistically significant result. It would also strengthen the conclusions to include an independent measure of presynaptic release probability that tests mHb to IPN synapses directly.

7) I am fundamentally confused about the logic of analyzing nicotine-induced responses in an animal that can't release Ach. Why would one look for or expect changes in a post-synaptic dependent mechanism given a pre-synaptic manipulation? Or is the logic that the nicotine induced behaviors all act via presynaptic receptors that regulate glutamate release?

*Reviewer #3:*

The paper does a good job of going from biophysics to behavior. As in all such studies, there are weaknesses; but on the whole the story is good.

Figure 1 and Figure 2: The mouse genetics, Cre recombinase, floxing, westerns, immunohistochemistry, and general anatomy look good. Figure 1: please move the LdT and BF labels, and use pointers, so we can see the areas better.

"This was expected, since cholinergic input from the stria medularis to the MHb (Contestabile and Fonnum, 1983; Gottesfeld and Jacobowitz, 1979) is not affected by conditional deletion of ChAT in the MHb" should be, "This was expected, since conditional deletion of ChAT in the MHb would not directly affect expression of any cholinergic genes in the stria medullaris, which gives rise to the presynaptic cholinergic input to the neurons studied. (Contestabile and Fonnum, 1983; Gottesfeld and Jacobowitz, 1979).”

The paragraph beginning "we next analyzed glutamatergic re-uptake" has some errors. More important is the conclusion that ChAT KO changed vesicular contents-the conclusion reached by comparing the change in mEPSC amplitudes vs. no change in GluR1 westerns. The conclusion is weak here, but subsequent data strengthen it. The westerns deal with both synaptic and extrasynaptic GluR1, of which a small number might be at the synapses of interest. Did perfusion with Ach or nicotine induce changes in holding currents?

Figure 3, title: change "Single" to “miniature".

Figure 4, Figure 5, and the associated paragraph: This sets up the analysis for Figure 5; the latter directly addresses the direct data for co-labeling.

Figure 5 attempts to co-label using different sized gold particles. Panel I has unhelpful horizontal axis label. Move I next to G, and use a common horizontal axis label. In the cartoon of H, make the gold particles different sizes.

More importantly, the analysis of colocalization is incomplete, because it lacks the controls: the authors don't show large-large and small-small gold particle distance distributions for comparison. Figure 4 shows that 40% of vesicles have VAChT particles; therefore if labeling is random, 0.4^2 = 16% of vesicles should show two VAChT particles. This should be true for 10% VGUT1 particles. You really should show this. These are underestimates assuming random labeling of vesicles.

Figure 6, using immuno-isolation, is good. Figure 6 gives the thermodynamic explanation that including ChAT activity makes VGLUT run better. The surrounding text of Figure 6 uses "rapidly", but there is no time course. To neuroscientists, "rapidly" can mean anything from microseconds to days. Please rephrase: "indicating that ACh via VAChT alkalizes SV, although considerably less than NH4". The next sentence, "The NH4 -induced effect was much more pronounced", is a hypothesis and should be stated, "possibly".

In Figure 7, the behavior is nice. The ChAT KO decreases locomotor inhibition and, perhaps, tolerance to this inhibition (this is limited by a ceiling effect). The KO decreases somatic signs of inhibition, anxiety, and CPP. The authors conclude that "elimination of cholinergic neurotransmission from MHb neurons is sufficient to abolish nicotine-induced responses including tolerance, withdrawal, and conditioned-reinforcement." This is fair, but it might not all be due to lack of vesicular synergy.

The Discussion is overall sensible, but the final hypothesis may go too far. You could also consider the possibility that eliminating ChAT decreases pH in the vesicle (increases deltapH), by decreasing the number of protons leaving. This, in turn, would increase nicotine accumulation in the vesicle (see Tischbirek, Neuron 2012). What might this do?

Finally, if the vesicle-loading hypothesis is correct, then the authors should phenocopy the acute electrophysiological effects with vesamicol.

---

## [Author Response]

*1) The assumption that two gold particles within a distance of 90 nm label the same vesicle is likely to give an overestimate of the number of vesicles containing both VAChT and VGLUT1, because gold particles on adjacent vesicles can easily be less than 90 nm apart given the close packing. The estimate of double-positive vesicles may be an overestimate, making it difficult to accept the authors' claims that the majority of SVs in these axon terminals contain both transporters. The authors should present a more compelling argument for why 90 nm is not only the maximum distance for labeling in the same vesicle but also the minimum distance for labeling in adjacent vesicles, or should make textual changes acknowledging the colocalization may be lower.*

We have edited this part in the manuscript (subsection “The vesicular transporters for acetylcholine and glutamate colocalize in the majority of synaptic vesicles of axonal terminals in the central IPN”) to indicate that the 90 nm maximal theoretical distance between two gold particles overestimates the number of double positive SVs because it does not exclude that two particles at a distance of less than 90 nm or even juxtaposed could be labeling adjacent vesicles. We have also made changes in Figure 6 to display the fraction of immunogold particles in distance distributions rather than categories.

*2) Given the similarity of the numbers at the bottom of subsection the “Cholinergic and glutamatergic inputs to IPN subnuclei” it would be appropriate to add the word "slightly", so that the sentence reads "slightly" less overlap in the IPI.*

We have edited this.

*3) The authors' claim that ChAT deletion resulted in the "complete absence" of nicotine-elicited behavioral responses is too strong of an assertion. Nicotine was still able to elicit hypolocomotion in the ChAT-cKO mice, albeit to a lesser degree than WT mice (Figure 11). The authors also did not provide any evidence to distinguish the effects of the loss of ACh release from these MHb neurons from the effects of altered Glu release from the same afferents. Therefore they are unable to convincingly argue that the behavioral effects are also due to the reported loss of the vesicle synergy and altered Glu signaling. Experiments that attempted to rescue impaired Glu release resulting from the loss of vesicle synergy (i.e. increasing Glu release from the ventral MHb-IPN projection of the ChAT-cKO mice) could have helped to dissociate the effects resulting from separate outcomes of ChAT deletion from ventral MHb neurons. The authors should make textual changes to moderate the assertion.*

We have edited this point in the manuscript (subsection “ChAT-cKO mice do not display nicotine-mediated reward and withdrawal behaviors”) and the figure legend (Figure 11) to remove “complete absence” and state that “these results reveal that nicotine-dependent behaviors such as nicotine-conditioned reward and withdrawal are controlled by the release of ACh from MHb neurons.”

[Editors’ note: the author responses to the previous round of peer review follow.]

As you had predicted, the additional experiments that were performed to address issues raised in review required several months to complete. We agree with the reviewers that additional data were required to strengthen the main conclusions of the paper, and we believe we have included all additional experiments that were requested in review. We have also extensively reorganized and revised the manuscript in response to the points raised.

*[…] Although you should address all of the points raised, the reviewers and editors agreed that the following experimental issues were the most essential: 1) The driver strain should be better characterized. At a minimum you should cross the Cre line to a reporter mouse and perform a Chat stain. This would allow you to then quantify both the penetrance and specificity of the line.*

We have included substantial further analyses of the driver mouse line A830010M20Rik-Cre crossed to the reporter mouse line Gt(ROSA)26Sor^tm1(EYFP)Cos^ in Figure 1 (new panels 1D-1E). Double immunostainings with CHAT and EYFP show that 99% (1912 of 1933) of CHAT positive neurons in the MHb are positive for the EYFP reporter while other CHAT populations in striatum, PPTg and LDTg show minimal expression of EYFP (0.5 to 1.3% of CHAT cells). These results demonstrate that the A830010M20Rik-Cre driver strain can be used to specifically target the cholinergic population of habenular neurons without affecting other cholinergic neurons (see subsection “Conditional gene deletion of ChAT in cholinergic habenular neurons”).

To assess the penetrance of the driver Cre-line, we provide quantification of the number of CHAT positive neurons in ChAT-cKO mice (A830010M20RikCre x floxed ChAT) in Figure 2 (new panels E and F). This analysis shows that only 0.3% habenular neurons are positive for CHAT, while the number of CHAT neurons in striatum, PPTg and LDTg are comparable in wt and ChAT-cKO mice. These data confirm that the A830010M20Rik-Cre strain drives Cre- recombination in 99.7% habenular cholinergic neurons (see the aforementioned subsection).

*2) You need to at least make a better argument for why non-habenular glutamatergic inputs are unlikely and may need to do some additional experiments to shore up this point.*

To clarify this point we have included a detailed explanation of the inputs to the IPN (subsection “Cholinergic and glutamatergic inputs to IPN subnuclei”). This section describes that the main glutamatergic input to the IPN is from the medial habenula, but there are other glutamatergic inputs to the IPN. These include projections from the laterodorsal tegmentum (LDTg), raphe nuclei, locus coeruleus, periaqueductal grey (PAG) and nucleus of the diagonal band. The neurotransmitter content of these other inputs is as follows:

a) The LDTg contains cholinergic, GABAergic and glutamatergic (VGLUT2) neurons (Geisler et al., 2007; Wang and Morales, 2009). Importantly the vast majority of CHAT positive neurons are not glutamatergic (Wang and Morales, 2009).

b) The raphe nuclei are mainly serotoninergic: only the dorsal raphe contains some glutamatergic neurons (VGLUT3) (Herzog et al., 2004; Jackson et al., 2009).

c) The locus coeruleus produces mainly norepinephrine, however some neurons may coexpress glutamate (Fung et al. 1994).

d) The PAG does not express VGLUT1, but does express a small amount of VGLUT2 (Geisler et al., 2007) and VGLUT3 (Herzog et al., 2004), and is very rich in GABAergic neurons (Reichling and Basbaum, 1991).

e) The diagonal band contains three populations of neurons: cholinergic, GABAergic and glutamatergic (VGLUT2 and VGLUT3 positive) (Geisler et al., 2007). Importantly none of the glutamatergic cells overlap with cholinergic markers (Henderson et al., 2010).

f) In addition the great majority of IPN neurons are primarily GABAergic excluding the possibility that the recorded mEPSCs are from local projections (reviewed in (Antolin- Fontes et al., 2015)).

Our experimental arguments supporting the conclusion that non-habenular glutamatergic inputs are unlikely are the following:

A) We provide new quantitation of the electron microscopy data that indicates that at least 80% (72 terminals out of 90) of synaptic terminals in the central part of the IPN (IPC) contain synaptic vesicles positive for both VGLUT1 and VACHT and that very few, 11%, are only glutamatergic (10/90)) (new Figure 6).

B) As a second method to quantify the degree of overlap of glutamatergic and cholinergic terminals in different nuclei of the IPN, we calculated the Manders’ colocalization coefficient that ranges from 0 for no colocalization to 1 for complete colocalization (new Figure 4) (Manders et al., 1993). These analyses indicate an extremely high colocalization of VGLUT1 and VACHT (M1=0.80) in the IPC and no colocalization in the lateral IPN (IPL).

C) As the published studies indicate (description of IPN inputs above), there are only three sources of cholinergic+glutamatergic projections to the IPN: the MHb, the LDTg and the nucleus of diagonal band. In both the LDTg and the nucleus of diagonal band, the neurons express either CHAT or VGLUT but not both markers in the same cell (Henderson et al., 2010; Wang and Morales, 2009).

D) Therefore in the IPC, where we have performed the EM studies and where the electrophysiological recordings show that glutamatergic mEPSCs differ in the ChAT- cKO, 80% of the inputs to the neurons we have recorded are glutamatergic+ cholinergic and thus they originate in the habenula.

E) In addition, we have performed electrophysiological recordings in the lateral IPN (new Figure 9), which receives non-cholinergic glutamatergic inputs, and we show no differences in glutamatergic mEPSCS between wt and ChAT-cKO. This supports the additional conclusion that glutamatergic inputs that are non-cholinergic are not affected by selective elimination of ACh in habenular neurons.

To clarify these points, we include new diagrams (Figure 4, Figure 5, Figure 7, Figure 8 and Figure 9) to indicate the specific subnuclei in the IPN where we have performed electron microscopy and electrophysiological recordings, and we have changed the order of the figures to introduce the EM findings and coefficient correlations before the electrophysiological recordings. Taken together, the published literature and our data argue strongly that the vast majority of cholinergic+ glutamatergic input to the IPC originates in the MHb.

*3) You need to better quantify and better support the western blot data, perhaps adding measures of holding current and paired pulse responses.*

We agree that this is an important point and have included more experimental data and edited this part in the manuscript. We agree that the information provided by the WB is not very quantitative since it does not distinguish between presynaptic, extrasynaptic and postsynaptic receptors and whether they are at the membrane or not. We have rephrased this part in the manuscript, we include additional electrophysiological measurements of holding currents and we have increased the sample size of mEPSC recordings upon nicotine and ACh bath application (Figure 8). These results show that the frequency of mEPSCs increases in wt but not in ChAT-cKO. This indicates that there is no presynaptic facilitation in the ChAT-cKO suggesting that the presynaptic nicotinic acetylcholine receptors (nAChR) in habenular terminals in the IPN are downregulated in the absence of ACh released by the terminals. In contrast, since the holding currents are similar between genotypes, postsynaptic nAChRs in IPN neurons are most likely not affected by elimination of ACh. Taken together, the new data in Figure 8 and the previous data in Figure 3 (now Figure 7) indicate that the function of presynaptic but not postsynaptic nAChR receptors is reduced in ChAT-cKO. The WB does not reflect this functional difference, indicating that the total number of nAChRs in the IPN is unchanged. Given the presynaptic effect we have measured, it is possible that ACh might be required for the trafficking of nAChRs to the membrane.

We have performed paired pulse responses by electrical stimulation of the fasciculus retroflexus (fr) and recordings in IPN neurons using oblique slices, which included the MHb, fr and IPN, but unfortunately there was too much variability between slices depending on the sectioning of the fasciculus retroflexus. Optogenetic recordings in ChAT-channelrhodopsin (ChR2) mice (as reported in (Ren et al., 2011)) would have been a better alternative, because these would allow direct stimulation of the MHb terminals, versus stimulation of the fr as is needed for electric stimulations. However in our case optogenetic recordings would have required crossing 3 mouse strains (Chat-ChR2 X A830010M20Rik-Cre X Chat _flox/flox_ and Chat ^flox/flox^ must be recombined in both chromosomes (5 generations, approx. 10 months). This is clearly beyond the scope of this revision, and we believe it should not be required.

*4) Additional controls are needed for co-localization experiments. This can probably be done simply by reanalyzing the images already acquired.*

As requested, we have re-analyzed the electron microscopy data and provided distance distributions of large-large (VGLUT1-VGLUT1) and small-small (VACHT-VACHT) gold particles in co-labeling EM experiments by quantitation of the number of gold particles which are at 0-90 (same vesicle), 90-180 (next vesicle) and more than 180 nm apart (Figure 6).

For VACHT-VACHT, the percentage of particles in the same vesicle is 61%. This is above random distribution (which would be 0.4_2_ =16%).

For VGLUT1-VGLUT1, the percentage of particles in the same vesicle is 36%. This is above random distribution (which would be 0.32_2_ =10%).

For VGLUT1-VACHT, the percentage of particles in the same vesicle is 54%. This is above random distribution (which would be 0.32 x 0.4 = 13%)

These values indicate that the double EM labelling of small-small, large-large and small-large particles is non-random and that the majority of SVs in the central IPN contain both vesicular transporters. We have indicated this in the Results section (subsection “The vesicular transporters for acetylcholine and glutamate colocalize in the majority of synaptic vesicles of axonal terminals in the central IPN”)

*5) The sample size for Figure 3 should be increased to ~20.*

We have increased the sample size to > 20 mice per genotype, and > 10 per condition. These data is now presented in Figure 8. The conclusions of these experiments have not changed, although the increased sample size has substantially strengthened them.

Reviewer #1:

*In this paper the authors use conditional deletion of CHAT to convincingly demonstrate synergy between ACh and glutamate at MHb synapses as well as showing changes in behavioral correlates of tolerance and withdrawal. These are important and interesting results. My major concern with the paper is that the authors have not demonstrated that this synergy is the only, or even a necessary component of mechanism of the behavioral changes they observe. Given the higher affinity of NMDA receptors than many other glutamate receptors, it is not clear that they would not be activated in the absence of facilitation of glutamate release. The issue may be somewhat semantic – that is the authors may wish "synergy" in the title and Discussion to apply more broadly to any sort of interaction between ACh and glutamate, as opposed to the vesicular synergy on which they focus for much of the results.*

The word synergy in the title is used in the broad sense described in the last sentence of this comment. We believe it is appropriately general for the *eLife* audience, although we would be happy to change the title if required.

*A western blot (Figure 3) is not a convincing way to demonstrate either that presynaptic receptors are unchanged or rule out a change in receptor number contributing to altered mini amplitude. If presynaptic receptors were altered this could also contribute to the altered frequency effects of added ACh and nicotine (3L and M).*

This was one of the major 5 points. We appreciate the insight from the reviewer and have included more experiments and edited this part in the Results and Discussion of the manuscript. See reply 3 of the summary points.

Reviewer #2:

*The study described in Frahm et al. investigates the mechanism and functions of glutamate/acetylcholine co-release from mHb inputs onto interpeduncular nucleus (IPN) neurons. The authors have an excellent model for the analysis of mHb neurons as the Cre line they describe may be specific for cholinergic neurons in the mHb. They go on to show that deletion of Chat from Cre+ neurons alters mEPSC amplitude in the IPN, and that VAChT and vGlut1 are both expressed on synaptic vesicles in terminals of the IPN. Finally, they show that this mouse line has an attenuated hypolocomotor response to nicotine injection and a lack of nicotine preference in a place preference task. While the authors offer some intriguing results there are major flaws in some of their conclusions. The study would also greatly benefit from a more thorough and quantitative characterization of their mHB/Chat cKO mouse model. 1) While the A830010M20Rik-Cre line may only express Cre in cholinergic neurons of the mHb, it clearly expresses Cre in many other areas of the brain. Quantification of the number of Chat+ neurons in the WT and Chat-cKO in the mHb is required at a minimum.*

This was one of the major 5 points. See reply 1 of the summary points.

*2) If nicotinic responses are still observable in IPN neurons when nicotine is directly applied (Figure 3), then the authors should also show that cholinergic responses from mHb are abolished in the IPN. An optogenetic experiment similar to that shown in Ren J., et al. 2011 would be appropriate.*

This was one of the major 5 points. See reply 3 of the summary points.

*3) While the authors show that mEPSC amplitude is decreased in IPN neurons of the Chat-cKO, mEPSC recordings sample from all input that a neuron receives, not just the inputs from mHb. The authors need to show that glutamatergic currents from mHB are affected in the Chat-cKO.*

See reply 2 of the summary points.

*4) In Figure 4 and Figure 5 the authors have no way of identifying if the terminals analyzed are from mHb or somewhere else, yet they state they are from the mHb. The wording should be changed to reflect this, or the authors should find a way to show that these terminals are indeed from the mHb.*

See reply 2 of the summary points. In addition we have changed “habenular terminals” for “axonal terminals” in the EM figures to reflect the fact that most presynaptic terminals but not all of them are habenular.

*5) The authors state that the quantity of vesicular proteins in WT and cKO are similar in Figure 6, however they do not quantify their results or perform any statistical comparison.*

We have quantified the levels of VACHT, VGLUT1 and VGLUT2 with respect to synaptophysin in wt and ChAT-cKO. These data are shown in Figure 10.

*6) The physiology experiments in Figure 3 are under sampled and require more cells in order to support the conclusions drawn from the statistically significant result. It would also strengthen the conclusions to include an independent measure of presynaptic release probability that tests mHb to IPN synapses directly.*

See reply 5 of the summary points indicating that the sample size was increased to > 20 mice per genotype, and > 10 per condition (wt: 25; cKO: 24, ACh: wt: 11; cKO: 12, Nic wt: 11; cKO: 12)

See also reply 3 of the summary points for the presynaptic release probability measurement.

*7) I am fundamentally confused about the logic of analyzing nicotine-induced responses in an animal that can't release Ach. Why would one look for or expect changes in a post-synaptic dependent mechanism given a pre-synaptic manipulation? Or is the logic that the nicotine induced behaviors all act via presynaptic receptors that regulate glutamate release?*

This is an interesting point. Given that nAChRs can act either presynaptically or postsynaptically, we were interested in the detailed mechanisms through which ACh acts when locally released in the CNS. Since the MHb is an extremely rich source of ACh, and since it is implicated in many different aspects of addictive behaviors, we performed this study to investigate the actions of ACh at these synapses and its effects on behaviour.

Reviewer #3:

*The paper does a good job of going from biophysics to behavior. As in all such studies, there are weaknesses; but on the whole the story is good. Figure 1 and Figure 2: The mouse genetics, Cre recombinase, floxing, westerns, and immunohistochemistry, and general anatomy look good. Figure 1: please move the LdT and BF labels, and use pointers, so we can see the areas better.*

We have made the suggested changes in new Figure 2 (old Figure 1).

*"This was expected, since cholinergic input from the stria medularis to the MHb (Contestabile and Fonnum, 1983; Gottesfeld and Jacobowitz, 1979) is not affected by conditional deletion of ChAT in the MHb" should be, "This was expected, since conditional deletion of ChAT in the MHb would not directly affect expression of any cholinergic genes in the stria medullaris, which gives rise to the presynaptic cholinergic input to the neurons studied. (Contestabile and Fonnum, 1983; Gottesfeld and Jacobowitz, 1979).”*

We have rephrased the sentence as suggested (subsection “Local elimination of ACh in medial habenular neurons reduces glutamate corelease”).

The paragraph beginning "we next analyzed glutamatergic re-uptake" has some errors. More important is the conclusion that ChAT KO changed vesicular contents-the conclusion reached by comparing the change in mEPSC amplitudes vs. no change in GluR1 westerns. The conclusion is weak here, but subsequent data strengthen it. The westerns deal with both synaptic and extrasynaptic GluR1, of which a small number might be at the synapses of interest. Did perfusion with Ach or nicotine induce changes in holding currents?

We appreciate the insight from the reviewer and have included more experiments and edited this part in the Results and Discussion of the manuscript. See reply 3 of the summary points.

*Figure 3, title: change "Single" to “miniature".*

We have changed this.

*Figure 4, Figure 5, and the associated paragraph:. This sets up the analysis for Figure 5; the latter directly addresses the direct data for co-labeling.*

We have now separated the electron microscopy Figure 4 and Figure 5 in a different manner as suggested by the reviewer: The single immunogold analyses are shown in Figure 5 which corresponds to the previous Figure 4 and Figure 5, and the double labeling analyses are presented in Figure 6 which corresponds to the previous Figure 5 plus 3 additional bar graphs.

*Figure 5 attempts to co-label using different sized gold particles. Panel I has unhelpful horizontal axis label. Move I next to G, and use a common horizontal axis label. In the cartoon of H, make the gold particles different sizes.*

We have made the suggested changes in Figure 5 (now Figure 6).

*More importantly, the analysis of colocalization is incomplete, because it lacks the controls: the authors don't show large-large and small-small gold particle distance distributions for comparison. Figure 4 shows that 40% of vesicles have VAChT particles; therefore if labeling is random, 0.4^2 = 16% of vesicles should show two VAChT particles. This should be true for 10% VGUT1 particles. You really should show this. These are underestimates assuming random labeling of vesicles.*

See reply 4 of the summary points.

Figure 6, using immuno-isolation, is good. Figure 6 gives the thermodynamic explanation that including ChAT activity makes VGLUT run better. The surrounding text of Figure 6 uses "rapidly", but there is no time course. To neuroscientists, "rapidly" can mean anything from microseconds to days. Please rephrase: "indicating that ACh via VAChT alkalizes SV, although considerably less than NH4". The next sentence, "The NH4 -induced effect was much more pronounced", is a hypothesis and should be stated, "possibly".

We have made the suggested changes and included the reference of a recent publication by two of the authors where the role of NH4+ in vesicular glutamate uptake is demonstrated (Preobraschenski et al., 2014).

*In Figure 7, the behavior is nice. The ChAT KO decreases locomotor inhibition and, perhaps, tolerance to this inhibition (this is limited by a ceiling effect). The KO decreases somatic signs of inhibition, anxiety, and CPP. The authors conclude that "elimination of cholinergic neurotransmission from MHb neurons is sufficient to abolish nicotine-induced responses including tolerance, withdrawal, and conditioned-reinforcement." This is fair, but it might not all be due to lack of vesicular synergy.*

We have changed this sentence in the manuscript (subsection “Absence of nicotine-mediated reward behavior and withdrawal in ChAT-cKO mice”).

*The Discussion is overall sensible, but the final hypothesis may go too far. You could also consider the possibility that eliminating ChAT decreases pH in the vesicle (increases deltapH), by decreasing the number of protons leaving. This, in turn, would increase nicotine accumulation in the vesicle (see Tischbirek, Neuron 2012). What might this do?*

We have changed the last part of the Discussion.

Generally the transport of each negatively charged anion like glutamate or Cl activates the proton pump to insert a proton for compensation. To sustain transport even if protons are accumulating, VGLUT itself then exchanges H+ against K+ as described in our recent paper (Preobraschenski et al., 2014). We carefully checked the literature and there is no indication that nicotine is actively transported over the plasma membrane by the choline transporter ChT (Mansner, 1977). We also found no indication that nicotine binds to VACHT or is actively taken up. So we think that even if nicotine will be distributed overall including glutamatergic terminal and vesicles, the effect seen will be on all glutamatergic vesicles/ synapses and should give an overall increase in mEPSC.

*Finally, if the vesicle-loading hypothesis is correct, then the authors should phenocopy the acute electrophysiological effects with vesamicol.*

We tried short application of vesamicol (VACHT blocker) in IPN neurons, but did not see any effects on mEPSC amplitude and/or frequency of mEPSCs in our initial experiments most likely because all the vesicles in the reserve pool have to be exocytosed before vesamicol can change the loading of the next set of SVs. Thus there is a problem with the experimental design that would require sustained activation of habenular neurons while applying TTX to measure mEPSCs. However, we think that these pharmacological experiments might not lead to further conclusions than the ones shown already with a cell specific genetic manipulation.